# NNsight and NDIF: Democratizing Access to Open-Weight Foundation Model Internals

**Jaden Fiotto-Kaufman,**[*] **Alexander R. Loftus,**[*] **Eric Todd, Jannik Brinkmann**[1,2]**, Koyena Pal,**
**Dmitrii Troitskii, Michael Ripa, Adam Belfki, Can Rager**[3]**, Caden Juang, Aaron Mueller,**
**Samuel Marks, Arnab Sen Sharma, Francesca Lucchetti, Nikhil Prakash,**
**Carla Brodley, Arjun Guha, Jonathan Bell, Byron C. Wallace, David Bau**
[1]Northeastern University, [2]TU Clausthal, [3]University of Hamburg

## Abstract

We introduce NNsight and NDIF, technologies that work in tandem to enable scientific study of very large neural networks. NNsight is an open-source system that extends `PyTorch` to introduce deferred remote execution. The National Deep Inference Fabric (NDIF) is a scalable inference service that executes NNsight requests, allowing users to share GPU resources and pretrained models. These technologies are enabled by the *intervention graph*, an architecture developed to decouple experimental design from model runtime. Together, this framework provides transparent and efficient access to the internals of deep neural networks such as very large language models (LLMs) without imposing the cost or complexity of hosting customized models individually. We conduct a quantitative survey of the machine learning literature that reveals a growing gap in the study of the internals of large-scale AI. We demonstrate the design and use of our framework to address this gap by enabling a range of research methods on huge models. Finally, we conduct benchmarks to compare performance with previous approaches. Code, documentation, and tutorials are available at `https://nnsight.net/`.

## 1 Introduction

Research on large-scale AI currently faces two practical challenges: *lack of transparent model access*, and *lack of adequate computational resources*. Model access is limited by the secrecy of state-of-the-art commercial model parameters (OpenAI et al., 2023; Anthropic, 2024; Gemini Team et al., 2024), and computational resources are limited by funding and engineering barriers. Commercial application programming interfaces (APIs) help amortize costs by offering frontier models "as a service". However, these APIs lack the transparency necessary to enable scientists to study model internals, e.g., by providing access to intermediate activations or gradients used during neural network inference and training.

Surveys of AI research needs have documented this situation (Shevlane, 2022; Casper et al., 2024). Bucknall & Trager (2023) highlight the need for *structured model access APIs* that offer greater transparency than existing commercial inference APIs. The current paper aims to meet this need by defining an expressive and efficient technical standard that addresses the key problems for providing a practical and useful structured model access API to support research on large models.

Our work makes three contributions:

**The *intervention graph* architecture** (Figure 1b), an approach for organizing experiments on very large models that reduces engineering burden, enhances reproducibility, and enables low-cost communication with remote models. We compare the intervention graph architecture to classical approaches for running similar experiments, and show how it achieves these goals.

**NNsight** (Figure 1a), an open-source implementation of the intervention graph architecture that extends `PyTorch` (Paszke et al., 2019) to create an expressive programming idiom that supports transparent model interventions on large-scale AI without requiring users to store or manage model parameters locally. NNsight builds intervention graphs using the same deferred computation graph

---

[*]Equal contribution. Correspondence to: {`j.fiotto-kaufman, loftus.al`}`@northeastern.edu`

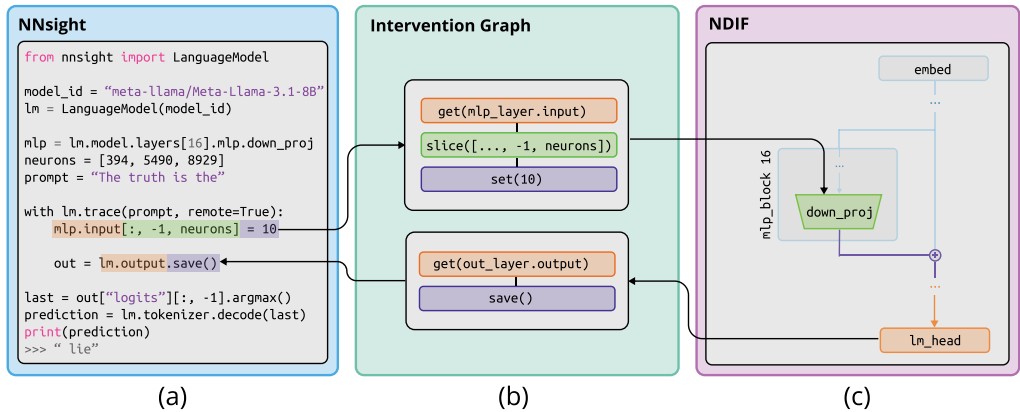

Figure 1: An example of the implementation of an NNsight intervention graph: (a) A user writes research code from which (b) an intervention graph is constructed. (c) The intervention operations are interleaved with the original model's computation and then executed. Values marked with `.save()` are made available to the user upon completion.

idiom that deep learning frameworks adopt to enable automatic differentiation (Bottou & Le Cun, 1988; Bottou & Gallinari, 1990; Abadi et al., 2016; Al-Rfou et al., 2016; Bradbury et al., 2018).

**NDIF** (Figure 1c), an open-source cloud inference service that supports this idiom by providing behind-the-scenes user sharing of model instances to reduce the costs of large-scale AI research. Crucially, by executing intervention graphs from multiple users with safe co-tenancy, NDIF supports the expressiveness of NNsight while avoiding costly startup times and communication costs incurred by traditional high-performance computing (HPC). We measure the performance of NDIF against the traditional approach, and we also compare it with Petals (Borzunov et al., 2023), a peer-to-peer approach to reducing barriers to distributed computing with large AI models.

## 2  SURVEYING MODEL AVAILABILITY AND RESEARCH USAGE

Compared to commercial APIs, models with openly downloadable parameters enable much more invasive experimentation and are a valuable resource for interpretability researchers (Scao et al., 2022; Biderman et al., 2023; Touvron et al., 2023; Jiang et al., 2023). To understand current research activity studying the internals of these open-weight models, we examine a set of 184 papers sourced from a recent survey of interpretability research (Ferrando et al., 2024) — see Appendix A for data curation details.

In Figure 2, we visualize the research usage of the largest open-weight models studied in these papers over time. While there has been a rapid investment in *training* larger and more capable models, we find that there is a disparity between the most capable systems[1] and those being *studied* in detail. There is a small group of papers that study language models with ≥70% MMLU performance (see (a) in Figure 2), suggesting that this gap can partially be explained as a previous lack of sufficiently capable models. However, even after the release of more-capable open-weight models, 60.6% of the surveyed research papers since February 2023 are still doing research on smaller, less performant models (< 40% MMLU). This suggests that the gap is not just due to a lack of capable models, but may also be due to engineering and infrastructural barriers.

Studying models with up to 70 billion parameters is possible for labs with access to high-end compute such as NVIDIA A100 or H100 80GB GPUs, but doing research *beyond* this scale represents a leap in technical complexity and resource requirements. For example, inference on a 530 billion parameter LLM uses 1TB of GPU memory, requiring many devices distributed across multiple nodes just to load model parameters in 16-bit precision (Aminabadi et al., 2022; Dettmers et al., 2023). Even if a scientist has access to such resources, they must integrate their experiment code with model sharding, parallelization, quantization, and other deployment specifics. Thus, despite evidence that

---

[1]Defined in terms of MMLU performance (Hendrycks et al., 2021, see Appendix A for details.)

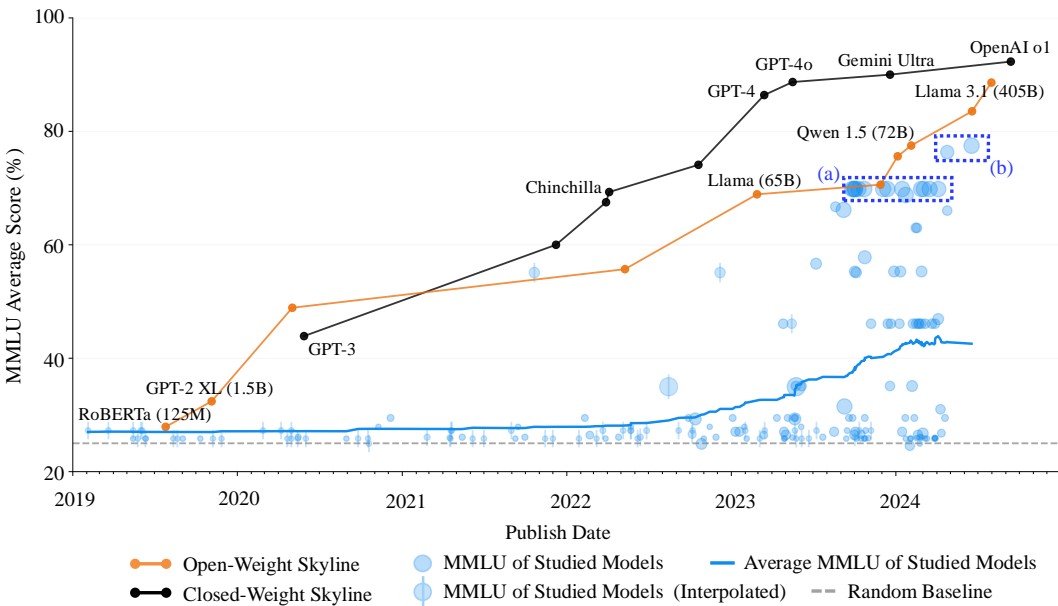

Figure 2: Most interpretability research is done on models that lag far behind the capabilities available in either closed- or open-models. Each blue point represents the MMLU performance of the largest open-weight model studied by a surveyed paper, where the size of the point represents the model's parameter size. Models without a recorded MMLU score were interpolated with nearest neighbors. There is a significant gap between the performance of models studied (blue line) and the capabilities of leading open-weight models (shown in orange). This gap is extended even further when considering the performance of leading closed-weight models (black line). (a) A small group of papers study language models with ≥70% MMLU performance, which can account for at least some of the gap. However, many researchers are still studying smaller, less performant models that hover around baseline performance (shown in gray). (b) Studying smaller, but still capable models such as Qwen 72B or Yi-34B may be part of the solution to closing this research gap, but these models still underperform the leading open-weight model, Llama 3.1 405B.

many fascinating capabilities appear only in the largest models (Brown et al., 2020; Rae et al., 2021; Wei et al., 2022; Patel & Pavlick, 2022; Schaeffer et al., 2023), studying them can seem unattainable.

One solution is to study similarly capable, smaller models; this approach is used in two recent works that study Yi-34B (Young et al., 2024) and Qwen 2 72B (Yang et al., 2024) (see (b) in Figure 2) — these are models that use distillation techniques to compress larger model capabilities into smaller model sizes. However, these models still underperform the most capable open-weight models, such as Llama 3.1 405B (Dubey et al., 2024). In the following sections we describe how NNsight and NDIF are designed to address this research gap by providing a flexible system for studying very large open models, presenting a reusable and modular service suitable for fully-transparent large-scale deployment distributed across many devices and nodes.

## 3 A FRAMEWORK FOR EXPERIMENTS ON LARGE-SCALE AI

Research on the internals of pretrained neural networks follows one common pattern: Experimental code is inserted between computational steps of the network, interleaved with code previously written to run the network. Depending on the experiment, researchers' code may inspect internal representations (Bau et al., 2017), perform interventions (Vig et al., 2020), collect gradients (Sundararajan et al., 2017; Selvaraju et al., 2017; Ancona et al., 2019; Bastings et al., 2022; Kramár et al., 2024), modify parameters (Meng et al., 2022), or use and train the parameters of supplementary models (Lester et al., 2021; Wu et al., 2023; Huben et al., 2024; Templeton et al., 2024; Marks et al., 2024).

Traditionally, researchers have organized their code by creating bespoke versions of neural network modifications and hooks for each experiment using callback systems (Li et al., 2024; Wang et al., 2022; Geva et al., 2023, and see Figure 3a), or by designing custom neural network systems with

built-in access points (Nanda & Bloom, 2022; Wu et al., 2024b). While tying experiments with network implementations is primarily a matter of code organization for small models, it burdens researchers with the responsibility and cost of model deployment and storage, increasing engineering challenges as parameter sizes increase. This pattern also makes it difficult for external research groups to reproduce large distributed neural network experiments without duplicating the environment.

We introduce a new framework that implements the following pattern: Separate experimental and engineering code, store the experiment, and then execute it on shared computational infrastructure. We achieve this separation by expressing an experiment as an *intervention graph* that can be saved, transmitted, optimized, and interleaved with model execution.

This framework implements three core requirements necessary for scalable, reproducible research that are not currently met by classical approaches:

- **Separation**: The framework must decouple experimental and engineering code, so that the increasingly complex challenge of running the underlying model can be tackled separately from experimental design.

- **Expressiveness**: The framework must allow the user to express all the experiments that they would have been able to express if they were using `PyTorch` on local hardware.

- **Co-Tenancy**: Experiments must be able to be safely, efficiently, and concurrently combined with model execution at runtime, to allow the service to amortize costs across many users.

Each of these requirements is addressed by one of the aspects of our framework: First, the *intervention graph* allows us to decouple research and engineering code. Next, the *NNsight API* is designed to be fully expressive, because users define interventions using `Python` and `PyTorch` code. Finally, the *NDIF inference server* enables co-tenancy, by setting up a central system to which experiments can be safely transmitted, optimized, and run concurrently.

## 3.1 REPRESENTING EXPERIMENTS AS GRAPHS

Our core innovation is a portable, serializable representation of an experiment, called the *intervention graph*, which dynamically captures modifications to the model's computation graph. This graph can be stored in `JSON` format, version-controlled, optimized, and sent to or retrieved from remote systems for execution. In this section, we formalize the notion of an intervention graph.

**Computation Graph.** Our formalism adopts Al-Rfou et al. (2016, Theano)'s definition of a computation graph[2]. The computation graph is a weakly connected, bipartite, directed, acyclic graph defined as $C = (V, A, E)$, where $V$ and $A$ are *variable nodes* and *apply nodes*, respectively. $E$ is a set of directed edges connecting the variable nodes and the apply nodes. The definition and properties of $V$, $A$, and $E$ are as follows:

- $V = \{v_1, \cdots, v_m\}$ is the set of variable nodes representing objects. They are one-to-many, and are connected to apply nodes.

- $A = \{a_1, \cdots, a_n\}$ is the set of apply nodes representing operations on variable nodes. They are many-to-one, and are connected to variable nodes.

- $E = \{e_1, \cdots, e_p\}$ is the set of directed edges. An edge $e$ in $E$ either connects a variable node $v$ to an apply node $a$, or vice versa. Thus, $E \subseteq (V \times A) \cup (A \times V)$ — making $C$ a bipartite graph as there are no connections between nodes of the same type.

In our context, a node that is "one-to-many" has a single input edge, but multiple output edges, as seen with variable nodes. Likewise, a node that is "many-to-one" has multiple input edges but only a single output edge, as with apply nodes. In NNsight, apply nodes correspond to `PyTorch Modules`.

The graph $C$ is *weakly connected* since converting all directed edges to undirected edges results in a connected graph. It is *bipartite* because edges only exist between variable and apply nodes.

---

[2]Al-Rfou et al. (2016) allow apply nodes to have many outgoing edges, whereas in ours, apply nodes can only have one outgoing edge. Appendix E argues that the two definitions are equivalent for our use case.

```
1  from transformers import AutoTokenizer, AutoModelForCausalLM        1  from nnsight import LanguageModel
2  model_id = "meta-llama/Meta-Llama-3.1-8B"                          2  model_id = "meta-llama/Meta-Llama-3.1-8B"
3  tokenizer = AutoTokenizer.from_pretrained(model_id)                3  lm = LanguageModel(model_id)
4  lm = AutoModelForCausalLM.from_pretrained(model_id)

5  mlp = lm.model.layers[16].mlp.down_proj                            4  mlp = lm.model.layers[16].mlp.down_proj
6  neurons = [394, 5490, 8929]                                        5  neurons = [394, 5490, 8929]
7  prompt = "The truth is the"                                        6  prompt = "The truth is the"

8  def pre_hook_fn(module, input):
9      input[0][:,-1,neurons] = 10

10 hook = mlp.register_forward_pre_hook(pre_hook_fn)                   7  with lm.trace(prompt, remote=True):
11 inputs = tokenizer(prompt, return_tensors="pt")                    8      mlp.input[:, -1, neurons] = 10
12 out = lm(**inputs)                                                 9      out = lm.output.save()
13 hook.remove()

14 last = out["logits"][:, -1].argmax()                              10  last = out["logits"][:, -1].argmax()
15 prediction = tokenizer.decode(last)                               11  prediction = lm.tokenizer.decode(last)
16 print(prediction)                                                 12  print(prediction)
                        (a)                                                                  (b)
```

Figure 3: Experiment code expressed using (a) standard `PyTorch` hooks and (b) the NNsight API. Both code snippets define the same intervention – activating three neurons which cause the model to invert the meaning of its output (e.g., producing "lie" rather than "truth"). The `PyTorch` intervention code captured in six lines of code (a, lines 8-13), can be easily expressed using NNsight with three lines of code (b, lines 7-9). Standard `PyTorch` requires creating custom hooks for each access point, whereas with NNsight, all module inputs and outputs can be accessed within a single `trace` context.

**Intervention Graph.** *Interventions* are modifications to the computation graph $C$ that introduce additional nodes and edges. We define an intervention component $C' = (V', A', E')$, where $C'$ is a computation graph representing a single user-specified edit. We define an intervention as $I = \{C', \mathcal{G}, \mathcal{S}\}$, where sets $\mathcal{G}$ and $\mathcal{S}$, referred to as *getters* and *setters*, determine how $C'$ connects to $C$. Integrating an intervention component into $C$ is called *interleaving*, and consists of two parts:

- The *getters* ($\mathcal{G}$) connect variable nodes in $C$ to apply nodes in $C'$, i.e., $\mathcal{G} \subseteq V \times A'$.

- The *setters* ($\mathcal{S}$) connect variable nodes in $C'$ back to apply nodes in $C$, i.e., $\mathcal{S} \subseteq V' \times A$.

The *intervention graph* of NNsight is the union of all individual intervention components, $\mathcal{I} = \bigcup_{1 \leq i \leq k} C_i'$. The final augmented computation graph is constructed by interleaving each intervention component with the original graph, and represents a complete user experiment. Appendix B.1 discusses implementation details of how intervention graphs are constructed, and Figure 10 (Appendix E) shows an example of an abstract intervention graph.

For the intervention graph to be valid, for every *getter* edge $(v_i, a_j') \in \mathcal{G}$ and *setter* edge $(v_k', a_l) \in \mathcal{S}$, there must be no directed path from $a_l$ to $v_i$ or from $v_k'$ to $a_j'$. This rule ensures that the augmented computation graph is acyclic.

## 3.2 Familiar and Expressive Interventions

To define and execute intervention graphs in code, the NNsight API uses `PyTorch` notation inside of a Python context window. We enable the tracing of user-defined experiments inside the context window by overriding basic Python operations as well as wrapping every `PyTorch` module and submodule to expose their inputs, outputs, and gradients. We call this the *tracing context*.

In Figure 3, we compare two code snippets that define the same intervention — one using standard `PyTorch` hooks (Figure 3a) and the other using NNsight's API (Figure 3b). In both snippets, a language model (`Llama-3.1-8B`; Dubey et al., 2024) is loaded from HuggingFace (Wolf et al., 2020) and an intervention is specified – activating three neurons which cause the model to invert its output (e.g., producing "lie" instead of "truth"). While using standard `PyTorch` hooks requires creating custom hooks for each access point (Figure 3a, lines 8-13), NNsight's `trace` context window provides access to the intermediate inputs and outputs of all `PyTorch` modules (Figure 3b, lines 7-9). See Appendix C for additional NNsight code examples.

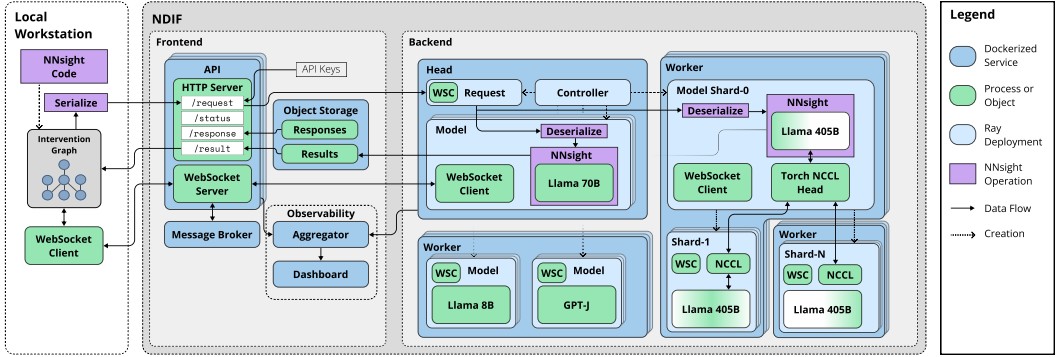

Figure 4: An overview of the NNsight and NDIF remote system. Researchers write experiment code using the NNsight API which is converted to an Intervention Graph. The graph is serialized to a custom JSON format and sent as a request to the NDIF frontend server. The NDIF backend can host multiple model instances, each on a dedicated set of GPU nodes. For very large models, such as Llama 3.1 405B (Dubey et al., 2024), model weights are distributed across many shards using tensor parallelism. The router transfers the request to the head node (shard 0) of the requested model, via the Ray GCS Service (Moritz et al., 2018; Ray Team, 2022). Shard 0 sends the request to all other shards of the model where it is then deserialized and executed. Each shard receives the full intervention graph, but only manages a slice of the model parameters. The Torch NCCL Head manages distributed model execution across allocated shards. After the intervention graph has been executed, results are gathered at shard 0 and sent to the object store in the NDIF frontend. The shard 0 WebSocket client informs the WebSocket client on the local workstation about the completion of the intervention. As soon as the local WebSocket client notes the intervention is complete it pulls the final results from the Object Store and inserts the result back into the local intervention graph. The research code can pull results from the intervention graph that are requested via .save().

The experiment's corresponding intervention graph is defined upon exiting the context window, and execution does not occur until the context is complete. The graph can then either be executed locally, or it can be serialized and sent to a remote system for execution. Tensors within the deferred execution block are not directly available to the main code, but every deferred tensor provides a .save() method to make its data available to the main program. Although the tracing context defers execution, it is still possible to debug many issues locally by using the PyTorch FakeTensor system, which precomputes and checks tensor shapes and datatypes while building the computation graph.

Because PyTorch code is used inside the tracing context, any experiment that can be written using standard PyTorch hooks can also be expressed using NNsight and will feel familiar to PyTorch users. NNsight wraps all 217 fundamental PyTorch tensor operations, supports built-in modules and custom neural network architectures, and includes infrastructure for loading models distributed by popular platforms such as HuggingFace. Users can express a wide range of interventions beyond simple probing and manipulation, including custom attention mechanisms, dynamic computation graph alterations, weight updates, and training loops (see Code Example 5), as well as computing gradients on backwards passes and gradient flow modifications.

By separating experiment code from the underlying neural network implementation, NNsight allows remote execution of experiments by simply adding a remote=True flag (Figure 3b, line 7), which sends the experiment to NDIF for execution.

### 3.3 CO-TENANCY AND REMOTE INFRASTRUCTURE

The NDIF service, in contrast to the NNsight system, is a multi-user runtime infrastructure designed to amortize costs and remove engineering burden from scientists by concurrently sharing inference. NDIF accepts intervention graphs created by NNsight, and executes them on large, preloaded models. In Figure 4, we provide an overview of the different components of the system, their responsibilities, and how they interact with one another.

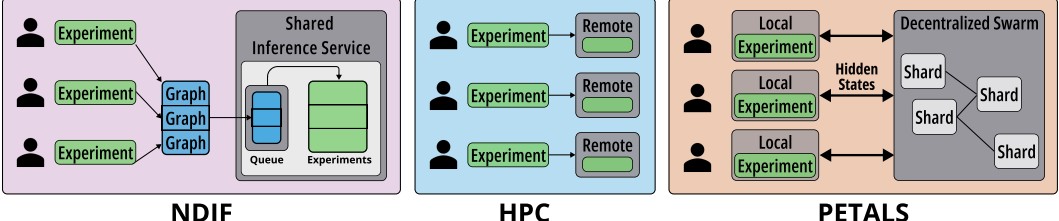

Figure 5: Schematic of research community use of NDIF vs. HPC and Petals. Green nodes show custom experiments. NDIF (left) allows many researchers to share a common inference service that runs customized experiments with shared memory. In HPC (center), researchers are responsible for weight-loading and handling model memory overhead on their own separate instances. In peer-to-peer swarm approaches like Petals (right), while GPU resources are shared, hidden states must be transferred between nodes during inference and returned to the user for custom interventions, resulting in costly data transfers.

Unlike traditional distributed computing approaches for supporting research experiments in AI, the NDIF infrastructure is designed to minimize the communication overhead involved in remote model execution. Figure 5 compares the infrastructure to traditional approaches.

In high-performance computing services (HPC), shared computational resources are allocated to researchers at the level of machines or virtual machines. That means that when researchers conduct experiments on customized AI models, all of the engineering code, weight-loading, and model storage must be handled by the researcher on instances that are exclusively allocated to them.

One approach that has been proposed for sharing costly computational resources while retaining researcher control is to distribute work using a peer-to-peer distributed service. This is the approach supported by Petals (Borzunov et al., 2023), which provides a swarm of inference servers that preload and provide inference computation services for layers of large neural networks. The Petals architecture can support custom experiment work by enabling researchers to host their own customized intervention code for particular layers on their own nodes (e.g., on their own workstation) while relying on the swarm for other inference steps. However, this architecture incurs costly data transfers multiple times during a forward pass for experiments that intervene on the inference process.

NDIF is designed to execute intervention graphs within the high-performance cluster itself. Models and engineering code are preloaded onto the cluster, and large model-internal data transfers are avoided since custom experiment code does not need to be run on the researchers' own hardware. Critically, unlike traditional HPC and peer-to-peer approaches, NDIF is designed to allow multiple researchers to conduct customized experiments while sharing the underlying preloaded model instances on shared computing hardware.

**Safe co-tenancy.** NDIF assumes responsibility for ensuring the safe and fair use of its shared model resources via multiple protocols. Interference between user experiments is prevented by the intervention graph's interleaving mechanism, which enables users to simulate interventions without permanently modifying the model's original computation graph. Model parameters cannot be directly accessed; instead, users interact with a copy of the parameters. Finally, users can only access models hosted on NDIF if they have been authorized by the model providers. This restriction is naturally enforced, as submitting a request to NDIF through the NNsight API requires downloading the model's meta version from Hugging Face, where access is managed directly by the model providers.

**Compute efficiency.** NDIF hosts a selection of preloaded models. For each model, a single shared instance is open to minimize performance difficulties associated with weight-loading and model startup. Researchers can access these models and conduct experiments on them through intervention requests conducted with the NNsight API. The infrastructure implements horizontal scaling and dynamic resource allocation to support multiple user requests and ensure the model deployments are operational and healthy at any given time. The NDIF server facilitates dynamic, bi-directional communication with the client side, enabling users to retrieve and save requested results, and allowing users to minimize additional overhead when running experiments remotely compared to locally. This approach reduces costs for individual researchers, as they no longer need to set up and maintain their

own separate high-performance computing environments. Figure 6 compares NDIF performance to using HPC and Petals.

The code for creating NDIF infrastructure is freely available on GitHub at `https://github.com/ndif-team/ndif`, allowing users to create their own NDIF infrastructure for specialized use cases.

## 4 PERFORMANCE AND EVALUATION

In this section, we evaluate the performance of NNsight when used remotely, both using exclusive allocations in the traditional high-performance computing setting, and using the shared NDIF inference service. We also compare to the Petals (Borzunov et al., 2023) distributed inference system, which can also reduce the costs of customized inference on huge models. Our evaluation focuses on the time required to load the model weights into memory, as well as the runtime of activation patching, a standard model intervention technique (Vig et al., 2020). For this purpose, we use a single batch of 32 examples from the Indirect Object Identification (IOI) dataset (Wang et al., 2022). We evaluate performance in three different settings: First, we measure the runtime when the models and interventions are executed entirely on a high-performance computing node (HPC). Then, we execute the models and interventions on the NDIF infrastructure, and contrast it against HPC execution times. Finally, we compare NDIF with Petals (Borzunov et al., 2023), and evaluate their relative performance on standard inference tasks as well as on intervention tasks.

**High-Performance Computing.** We first evaluate NNsight in HPC scenarios by comparing it against other popular libraries designed for intervening on the internal states of `PyTorch` models, namely `TransformerLens` (Nanda & Bloom, 2022), pyvene (Wu et al., 2024b), and baukit (Bau, 2022). Our comparison spans models ranging from 1.5 to 8.5 billion parameters. We consider GPT2-XL (Radford et al., 2019), Gemma-7B (Team et al., 2024), and Llama-3.1 8B (AI@Meta, 2024). The experiments were conducted on a single HPC node with four NVIDIA H100 PCIe 82GB GPUs with CUDA version 12.3 and an Intel(R) Xeon(R) Gold 6342 CPU with 24 cores.

The results are presented in Table 1. We find that the other libraries achieve comparable weight-loading and activation patching times[3] to NNsight run without NDIF, for all three models. Therefore we argue that when using exclusive HPC allocations, the choice of library should primarily be guided by factors such as usability, feature set, and compatibility with existing workflows.

Table 1: Runtime comparison of baukit, pyvene, and `TransformerLens` with NNsight for loading `PyTorch` models into memory and intervening on their internal states. NNsight achieves comparable performance with other frameworks in both evaluation settings across all three models.

| Framework | Setup Time | | | Activation Patching | | |
|---|---|---|---|---|---|---|
| | GPT2-XL | Gemma-7B | Llama-3.1-8B | GPT2-XL | Gemma-7B | Llama-3.1-8B |
| baukit | $3.146 \pm 0.006$ | $6.112 \pm 0.074$ | $4.529 \pm 0.049$ | $0.073 \pm 0.001$ | $0.220 \pm 0.006$ | $0.352 \pm 0.006$ |
| pyvene | $3.143 \pm 0.011$ | $6.263 \pm 0.487$ | $5.736 \pm 0.717$ | $0.074 \pm 0.001$ | $0.222 \pm 0.001$ | $0.348 \pm 0.006$ |
| TransformerLens | $8.991 \pm 0.024$ | $21.560 \pm 0.385$ | $20.625 \pm 0.365$ | $0.205 \pm 0.027$ | $0.208 \pm 0.002$ | $0.332 \pm 0.002$ |
| NNsight | $3.532 \pm 0.172$ | $6.351 \pm 0.062$ | $5.991 \pm 0.192$ | $0.073 \pm 0.002$ | $0.227 \pm 0.009$ | $0.335 \pm 0.010$ |

**Remote Execution.** We measure the same evaluation settings for HPC versus remotely on an NDIF server. We compare the two setups using the OPT suite of language models (Zhang et al., 2022), ranging from 125 million to 66 billion parameters. For the experiments, we use the same infrastructure as in the comparison against other libraries. The results of this analysis are presented in Figure 6a & 6b. As model size increases, the time required to load the model into memory grows nearly linearly on HPC. In contrast, the time for loading is negligible when using NDIF, since models are preloaded by the service. We also observe that the remote execution on NDIF introduces a roughly constant overhead from communication between the local device and NDIF during activation patching, regardless of model size (see Table 2 in Appendix D). In our setting, remote execution provides runtime improvements for models with 3 billion parameters and more, indicating that remote execution via NDIF becomes increasingly beneficial as parameter size grows.

---

[3]There is one exception, as loading the model weights into memory takes about three times as long with `TransformerLens` as with other libraries. This is likely a result of the preprocessing steps to convert weights into a standardized format across different models.

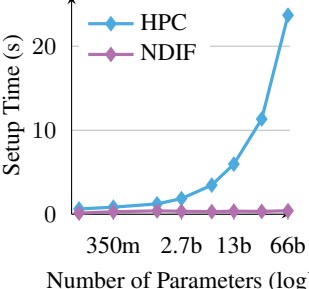 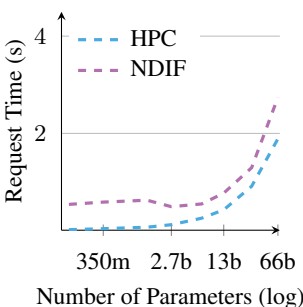 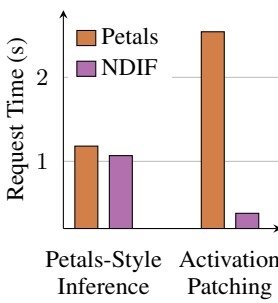

(a) **HPC vs. NDIF.** We compare the time required to set up the model when NNsight is executed on an HPC node and an NDIF server. For HPC execution, the model must be loaded, so the setup time scales nearly linearly with model size. In contrast, remote execution with NDIF bypasses loading weights.

(b) **HPC vs. NDIF.** We also compare the runtime of activation patching when NNsight is executed on an HPC node, versus using a NNsight locally to remotely access an NDIF server. We observe that the remote execution introduces a roughly constant overhead from communication between the local device and NDIF.

(c) **Petals vs. NDIF.** We compare NDIF against Petals, an open-source remote inference framework. We find that the libraries demonstrate comparable performance for standard remote inference, but that NDIF significantly outperforms Petals for remote interventions.

Figure 6: Performance evaluation of NNsight and NDIF. In all experiments, the sample size is $n = 128$. Plot marker widths are comparable to the 95% confidence intervals.

**Distributed inference.** We also compare NDIF against Petals, an open-source service for distributed model inference (Borzunov et al., 2023). Petals enables inference on large language models across shared resources by transmitting intermediate hidden states. To ensure a fair comparison, we deployed private instances of both Petals and NDIF on a server with a single NVIDIA RTX A6000 49 GB GPUs with CUDA version 12.5 and an AMD Ryzen 9 5900X CPU with 12 cores. The communication between these instances took place via a network with a bandwidth of about 60 MB/s. As Petals was primarily designed for standard remote inference, we first compare the two services in this context. With Petals, the client transmits token embeddings to the server and receives final hidden states, which can then be mapped to token probabilities. In contrast, NDIF sends the input data and the intervention graph to the server. While NDIF could simply return the next token prediction, we opted to return the final hidden states to the client for a fair comparison. The results are presented in Figure 6c. In this scenario, both frameworks demonstrate comparable performance.

Petals also support certain types of inference-time interventions on model internals. Therefore, we also compare NDIF and Petals in executing activation patching. Petals allows returning intermediate hidden states, which can be used for interventions on the model's residual stream. However, as Petals does not support server-side interventions, modifications to the internal state must be performed locally on the client device. This process involves receiving hidden states at a specific layer, performing local modifications, and then sending the modified hidden states back to the server. In contrast, NDIF supports server-side interventions and metric computations. This allows us to avoid transmitting hidden states between client and server by computing patching metrics (e.g., logit difference) on the server and returning only those. The results are presented in Figure 6c. In this scenario, we find that NDIF significantly outperforms Petals.

## 5 RELATED WORK

**Research model hosting frameworks.** The most similar previous works are Garçon (Elhage et al., 2021) and Petals (Borzunov et al., 2023). Garçon is a proprietary internal research system at Anthropic that supports remote inspection and customization of large models hosted on a remote cluster. Unlike the computation-graph approach of NNsight, Garçon operates by allowing researchers to hook models via arbitrary-code execution. As a result, co-tenancy is unsafe, so unlike the NDIF server that shares model instances between many users, Garçon requires each user to allocate their own model instance, which is computationally expensive. The Petals system has similar goals, but instead of distributing experiment code, it distributes foundation-model computation across user-contributed nodes in a BitTorrent-style swarm. Petals achieves co-tenancy by leaving researcher-defined computations on

the user's own system. Unlike Petals, NNsight can avoid costly model-internal data transfers by executing computation graphs on NDIF servers.

Several other solutions for sharing hosted model resources have been developed, including VLLM (Kwon et al., 2023), Huggingface TGI (Dehaene et al., 2024) and Nvidia TensorRT-LLM (Nvidia, 2024). These are systems for LLM inference acceleration that support large-scale multiuser model sharing. However, unlike NDIF, these systems only provide black-box API access to models, and do not permit model inspection or modification, which limits their utility for interpretability research. Also related to our work are systems such as S-LoRA (Sheng et al., 2024), dLoRA (Wu et al., 2024a), and Punica (Chen et al., 2024), which excel at efficient serving of many pretrained LoRA adapters in parallel, and systems such as FusionAI and FusionLLM (Tang et al., 2023; 2024) which support decentralized training. While these LoRA-focused systems enable one form of model modification, NNsight and NDIF provide finer-grained control that allows for a wider range of model interventions.

**Frameworks for model internals.** Numerous efforts have been made to create robust tools for exploring and manipulating model internals. These tools are all designed to support *local* experiments, executed on compute resources controlled by the researcher conducting them. `TransformerLens` (Nanda & Bloom, 2022) is designed to facilitate the exploration of GPT-style decoder-only language models. It allows users to apply custom functions to outputs of specific model modules. Unlike NNsight, which operates on arbitrary `PyTorch` networks, `TransformerLens` is limited to transformer-based language models. Pyvene (Wu et al., 2024b) supports configurable intervention schemes on `PyTorch` modules, decorating a `PyTorch` module with hooks that allow activations to be collected and overwritten. Pyvene operates at a higher level of abstraction, and can be configured as a layer over NNsight. `Baukit` (Bau, 2022) provides a range of utilities that simplify tracing and editing activations for local models. However, unlike NNsight's intervention graph, it lacks an intermediate representation that would allow for interaction with larger, remotely-hosted models. `Penzai` (Johnson, 2024) is an open-source functional library which provides a modular system that allows for introspection, visualization, and manipulation of neural network computation graphs; `Penzai` works within the JAX framework, in contrast to NNsight which works with `PyTorch` models.

## 6 DISCUSSION

The growing scale of highly-capable AI systems has presented our field with an access challenge: The most capable systems are increasingly costly to deploy, and that has meant that many of the most interesting and important research targets have become difficult for scientists to study in depth. In this paper we have proposed an architecture for reducing barriers to research by defining a technical approach for separating experiment definition from network deployment. Together, the intervention graph architecture, the NNsight API, and the NDIF service enable researchers to perform complex experiments and explore neural network internals at a scale previously limited by computational, engineering, and financial constraints.

While our approach addresses technical issues for conducting research on very large open-weight models, a key limitation is that it does not provide access to closed-parameter, proprietary models like GPT-4 or Claude. However, proprietary vendors often have special requirements. For instance, they may wish to maintain secrecy and security of model parameters for business or safety reasons. Our system is designed with these concerns in mind, and it is technically feasible for companies to support under these requirements. We encourage commercial providers to consider adopting our approach as part of their products to enable transparency and facilitate future scientific progress at the frontiers of AI.

NDIF and NNsight define a way of doing research that acknowledges the important role of community resources. Scientists in other fields have opened up new research vistas by pooling investments in shared research instruments, databases, and facilities. By enabling the AI research community to similarly pool its resources for large-scale inference, we can open a pathway for conducting research on AI at scales that would be unattainable for individual research groups.

## 7 ETHICS STATEMENT

Our work is intended to democratize access to large neural networks by providing tools that make it easier for researchers to audit, understand, and interpret large AI models. We caution that such transparency may also enable abuse of large models, for example exploiting model vulnerabilities. Therefore, we strongly urge the responsible and ethical use, deployment, and monitoring of our tools, while emphasizing the importance of continued research to improve model interpretation.

## 8 REPRODUCIBILITY STATEMENT

To promote transparency and ensure the reproducibility of our results, the complete NNsight and NDIF frameworks, along with comprehensive documentation and all relevant experiment code, are available at `https://github.com/ndif-team/nnsight` and `https://github.com/ndif-team/ndif`, respectively. We hope this will allow researchers and practitioners to build upon our work.

## 9 ACKNOWLEDGEMENTS

The National Deep Inference Fabric (NDIF) and NNsight are supported by the U.S. National Science Foundation Award IIS-2408455. We are also grateful for the support of Open Philanthropy (ET, KP, SM, AS, NP, DB), Manifund Regrants (CR), and the Zuckerman Postdoctoral Fellowship (AM). We thank Brett Bode, William Gropp, Greg Bauer, and Volodymyr Kindratenko at NCSA for their help working with DeltaAI, which provides the primary computing infrastructure for NDIF, and we thank Thea Shaheen for invaluable assistance with infrastructure for the project at Northeastern. We thank the New England Research Cloud (NERC) for hosting the web frontend for NDIF. We also thank Callum McDougall for developing the NNsight tutorial in the ARENA 3.0 materials. We thank Clément Dumas and Gonçalo Paulo for valuable feedback as early research adopters, Zhengxuan Wu for implementing pyvene on NNsight, and Zhongbo Zhu for valuable advice on serving infrastructure. We also acknowledge the many early testers for the Llama 3.1 405B pilot program. Finally, we are grateful for the ongoing feedback and engagement of our community of users on Discord.

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

# A    RESEARCH SURVEY DETAILS

In this section, we provide details on the data collection process for the results reported in Section 2, and report additional findings related to model parameter size.

**Data Curation**. We curated a list of 184 total papers that study the internals of open-weight transformer models, taken from the citations of a recent survey of interpretability research (Ferrando et al., 2024). Starting from the initial list of 411 citations, we excluded any paper that met one or more of the following criteria:

- Paper is a survey paper, tutorial, library report, model card, or performs no model experiments.
- Paper only performs experiments on closed-weight models, custom models, or non-transformer architectures.
- Paper was published before 2019.

We identified the largest model (by parameter count) studied in each paper and gathered data related to its MMLU performance  (Hendrycks et al., 2021) from the HuggingFace OpenLLM Leaderboard Archive (Huggingface, 2024). We used the Papers with Code MMLU Leaderboard (PaperswithCode, 2024) to supplement MMLU performance data when models did not have a score on the Huggingface leaderboard. When a model had a score from both sources, we averaged its benchmark results together. A few models did not have reported MMLU results from either source (primarily BERT-style models). For these models, we interpolated MMLU results using nearest neighbor approximation based on parameter size. Papers with Code provides an MMLU performance reference for closed-weight models.

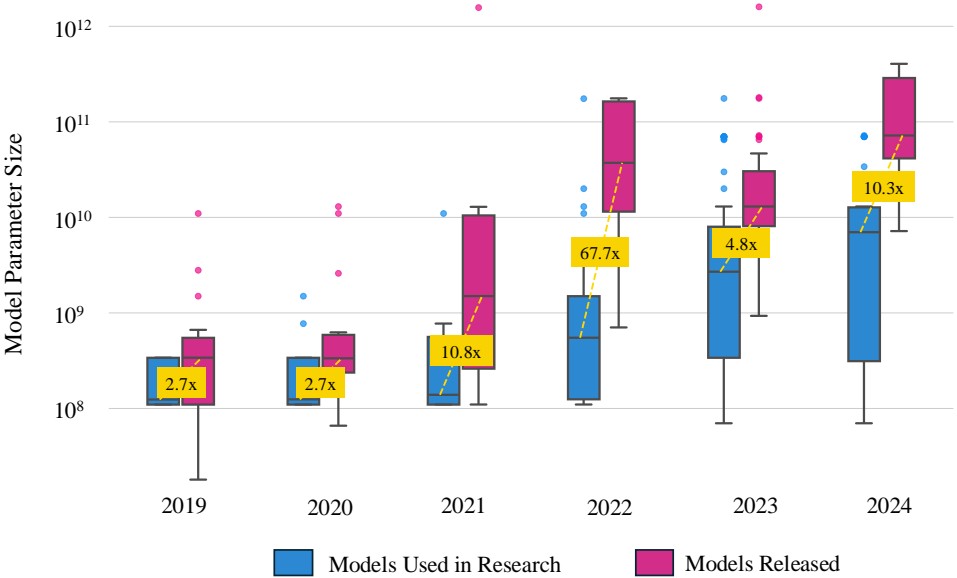

Figure 7: Evolution of the size gap between open-weight models used in research and publicly released models from 2019 to 2024. Blue boxes represent the distribution of model sizes used in research papers, while pink boxes show publicly released models. The dashed gold line connects median model sizes for each group, with the ratio between medians displayed. Outliers are shown as individual points. The growing disparity (from 2.7x in 2019-2020 to 10.3x in 2024) suggests that interpretability research is increasingly lagging behind the capabilities of state-of-the-art models available to the public.

**Comparing Model Parameter Size.** In Figure 7, we show there is a growing disparity between the size of models that interpretability researchers are studying and the size of models that are publicly released each year, a similar trend to the one discussed in §2 regarding MMLU. Reference data on parameter sizes of publicly available models is taken from Epoch AI (2024), restricting our analysis to language models released after 2019 that have openly downloadable parameters. To ensure reproducibility and transparency, all source code and data used for the analyses described in this section are available in the supplementary materials accompanying this paper.

## B SYSTEM DESIGN

In this section, we provide additional details describing the implementation and design of NNsight and NDIF.

### B.1 NNSIGHT DESIGN

**Wrapped PyTorch modules**    To support the tracing context, the NNsight library wraps PyTorch modules in an NNsight object instance. This wrapper is responsible for providing access to its sub-modules' intermediate deferred inputs and outputs. Upon initialization, the NNsight object creates an Envoy object for each sub-module, forming a tree-like structure mirroring that of the original PyTorch module tree. Each Envoy is responsible for managing and recording operations on future inputs and outputs for its underlying module. It achieves this by exposing attributes like .input and .output which, when accessed, trace a deferred operation to intervene at the corresponding module's input or output during model execution. This attribute access returns a Proxy object of that deferred operation, representing a future intermediate value. Any operation performed on the resulting Proxy object creates a new deferred operation, and therefore a new Proxy. As all future Proxys originate from operations performed on these root .input and .output attributes, it makes them the entry-point into the tracing context functionality.

```
1 from nnsight import NNsight
2 model = NNsight(model)
3 input = tensor([.46, 1.56, -0.3, .98, -0.5])
4 with model.trace(input):
5     model.submodule.output[:2] = 0
6     hidden_state = model.submodule.output.save()
```

Code Example 1: Basic usage of the NNsight tracing context and the NNsight data type. model in this example has a single submodule named "submodule" and accepts a Tensor of length five. On line two, the PyTorch model is wrapped with NNsight. On line four, the tracing context is created and entered by calling with model.trace(...) given some input. On line five, the Envoy for "submodule" is used to access a Proxy for its deferred intermediate output via the .output attribute. Also on line five, the returned Proxy is intervened on, creating two new Proxys and Nodes through the overloaded slicing and setting methods. Finally on line six, the same intermediate value is placed in the "hidden_state" variable, and saved for use after model execution.

When exiting the tracing context, the NNsight object interleaves the traced operations to the model's computation graph and executes the model. This process is performed by adding PyTorch hooks to all modules whose Envoy's .input or .output attributes were accessed within the tracing context. These hooks are the interface between the intervention graph, formed from the deferred operations gathered while inside of the tracing context, and the model's original computation graph. After interleaving, NNsight removes these hooks.

**The Intervention Graph**    The Intervention Graph encapsulates the logic that should be interleaved with model's original computation graph. The Graph is composed of individual Node objects, each representing a single operation to be executed during evaluation of the graph. Nodes have arbitrary positional and keyword arguments. These arguments may contain other Nodes which are considered *dependencies* of the source Node. Conversely, the source Node is a *listener* of its *dependency*. When a Node's remaining *listeners* reaches zero, it is considered redundant and has its value is cleared and its memory freed immediately.

Portions of the Intervention Graph are divided into sub-graphs depending on which module they rely on. These sub-graphs are then executed sequentially when their corresponding module's hook is called during model execution. In other words, the root intervention Nodes created via Envoys (.input or .output) act as GOTO statements that transfer execution of the Intervention Graph, beginning from that Node up until they hit another root intervention Node. Modules may be executed more than once during interleaving, and therefore these sub-graphs may also be executed multiple times, just as you would a for loop.

For every Node, there is a corresponding Proxy that manages tracing operations on the underlying Node. Proxys achieve this by overriding builtin Python and PyTorch magic methods. These overrides

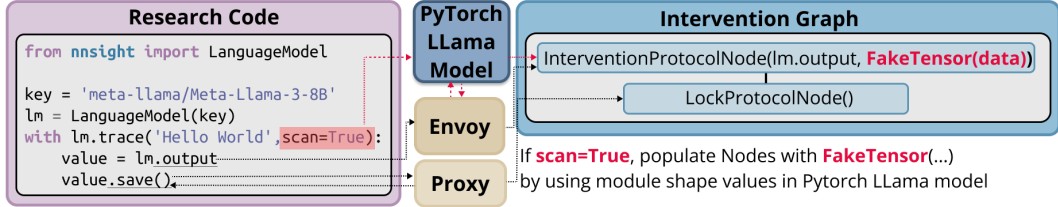

Figure 8: The components involved in creating nodes within the Intervention Graph. Module calls like `lm.output` are processed through `Envoy`, resulting in the creation of relevant `ProtocolNodes` like `InterventionProtocolNode()`. Everything else goes through `Proxy` layers to create other relevant nodes in the graph. All nodes are encapsulated within `InterventionProxy()`, which users see upon printing nodes. NNsight can scan the modules in the `PyTorch` model instance to gather related data shapes with a `scan` flag, creating `FakeTensors` that are included in the respective intervention graph nodes and returned to the user.

intercept calls to methods and functions that now create and add a new `Node` to the `Intervention Graph`, returning the `Node`'s corresponding `Proxy`. Chaining this process results in a completed `Intervention Graph`.

**Protocols**   When composed into an `Intervention Graph`, `Nodes` are quite powerful on their own. However, a full-featured framework requires operations that can exert more control on the intervention process than the scope an individual `Node` can provide.

A `Protocol` is a special NNsight operation that enables these more powerful controls. A `Protocol` must define two methods: One that adds a `Node` of its type to the `Intervention Graph`, and another that executes that `Node` during interleaving. Rather than NNsight executing a `Node`'s target with its arguments, `Protocols` allow the `Node` to be passed the `Protocol` for execution. With full access to the `Node`, the `Protocol` can processes its arguments, implement complex logic, and access the `Intervention Graph` itself. This access provides the `Protocol` a global view of the intervention process, which it can leverage by interacting with the `attachments` attribute on the `Intervention Graph`. `Attachments` are a dictionary on the `Intervention Graph` where `Protocols` can manage a global state throughout interleaving.

Two types of `Protocols` are highlighted below:

`LockProtocol`: When a `Node` has no more listeners, its value is cleared to free memory. Therefore this NO-OP operation never updates the remaining listeners of its dependencies, preventing a `Node`'s value from being deleted. `Proxys` implement a `.save()` method which creates a `LockProtocol` with the source `Node` as its only dependency, resulting in the `Node`'s value being available outside the tracing context.

`GradProtocol`: Many use cases require intervening on the backwards pass of a model rather than exclusively the forward pass. Accessing a `Proxy`'s `.grad` attribute creates a `GradProtocol` `Node`. When executed, `GradProtocol` adds a backwards hook to the underlying `Tensor` value. This hook injects the gradients into the `GradProtocol` `Node` during the backwards pass, allowing it to behave like any other `Node`.

**Model Abstractions**   The `NNsight` class defines abstract convenience methods that subclasses can implement to extend common neural network architectures to the NNsight API. These methods include logic to load the model, which is called twice when initializing and executing an NNsight-based model from scratch. The first call happens upon initialization to create and populate the `Envoy` tree with a 'meta' version of the model. In a 'meta' model, the parameters of the network are not yet loaded: they exist as placeholders containing shape and datatype information. The second call occurs when loading and dispatching the true model parameters.

NNsight comes with a few of these subclasses pre-implemented. The most common subclass is the `LanguageModel`, for text-to-text sequence generation architectures. The `LanguageModel` integrates with `Huggingface` to load model configurations, tokenizers, and parameters from their online repository when provided with a repo id. Implementing the input processing methods uses

the model's tokenizer to accept common forms of inputs and batch them together. When loading the 'meta' version of the model, the LanguageModel uses AutoModelForCausalLM.from_config from Huggingface transformers to download only the models configuration. Only upon dispatching does LanguageModel fetch the model parameters if they aren't already cached.

**Scanning and Validation**    NNsight offers two key debugging features integrated with the Tracer context.

The first is Scanning. When toggled, scanning performs an initial execution pass on the model in FakeTensorMode. During this pass, Hooks registered on each module update the input and output of their corresponding Envoys on the NNsight model with generated FakeTensors. This process makes valuable information available during tracing such as output shapes, dtypes, and device allocations.

When Scanning is enabled, Validation can be used to test interventions on FakeTensors as they are being added on the intervention graph. This acts as a sanity check to identify errors related to defined interventions before engaging in the model's forward pass run.

**Remote Execution and Session**    It is common for users to run multiple forward passes on a model as part of a larger experiment, such as in a LoRA training loop. NNsight offers an API that allows the creation of a Session context, where inner defined Tracer contexts are executed sequentially only after the Session context is exited. By encapsulating multiple Tracers in a Session, values obtained in earlier passes can be seamlessly referenced by later stages in the experiment without the need for manual saving.

When a Session is executed remotely, all forward passes defined within are sent as part of a single request to be executed. This can reduce wait times between Tracer calls by 1) minimizing the number of server requests and 2) eliminating the need to download and re-upload values referenced across Tracer contexts for subsequent calls that require them.

## B.2    NDIF DESIGN

NDIF enables distributed execution of intervention graphs across multiple GPUs. The system architecture consists of three main components: a request handling layer, a model service layer, and a distributed execution engine. The current implementation builds on top of Ray (Moritz et al., 2018; Ray Team, 2022) as a distributed scheduler.

**Request Processing and Graph Distribution**    When a user creates an intervention with NNsight, the system serializes the intervention graph into a custom JSON format and sends it to NDIF's HTTP server front-end. The request handler places the graph in Ray's object store and queues it for processing by the Request Service deployment. This service routes the request to the appropriate Model Service deployment based on the requested model.

For distributed models split across multiple GPUs, NDIF maintains separate Ray deployments for each model shard. The head shard (shard 0) receives the request first and distributes it to other shard deployments. All shards wait to receive the complete intervention graph before beginning execution.

**Distributed Model Management**    NDIF loads large models using a sharding process. First, it creates a "meta model" containing only shape information without loading weights. It then attaches hooks to PyTorch's load_state_dict functionality that convert modules into their parallel versions during loading. This allows NDIF to load model checkpoints shard-by-shard while maintaining a set memory limit for unsharded tensors at any given time.

**Executing Distributed Interventions**    During execution, NDIF handles interactions between the distributed model and the intervention graph. When encountering distributed tensors (DTensors), NDIF records tensor placement information, converts DTensors to full tensors using torch.distributed gather operations, injects the full tensors into the intervention graph, and then re-shards tensors after graph execution using saved placement information.

This approach allows the intervention graph to operate on complete tensors while maintaining memory efficiency. Only the head shard handles communication with the user, including websocket updates and returning saved values.

**Co-tenancy and Resource Management**   NDIF supports sequential co-tenancy, allowing multiple users to access the same model instance without reloading parameters. Future implementations will enable parallel co-tenancy through batch grouping as follows: During tracing, intervention nodes record batch groups that specify tensor slices. During execution, the system extracts appropriate slices while preserving gradient propagation, enabling multiple users to share execution within a single forward pass.

## C CODE COMPARISON

In Section 3.2, we discussed how NNsight can express any experiment that can be expressed with PyTorch, but that is also simplifies the design of custom experiments on large model internals compared to using standard PyTorch hooks. Here we provide an additional pair of code examples that implement a common interpretability technique called "activation patching" (Zhang & Nanda, 2024). Code Example 2 implements this experiment using PyTorch hooks, while Code Example 3 uses the NNsight API. Both code snippets implement the same intervention, causing the model to change its prediction for the base prompt from "Paris" to "Rome".

```python
from transformers import AutoTokenizer, AutoModelForCausalLM
model_id = "meta-llama/Meta-Llama-3.1-8B"
tokenizer = AutoTokenizer.from_pretrained(model_id, padding_side='left')
tokenizer.pad_token = tokenizer.eos_token
lm = AutoModelForCausalLM.from_pretrained(model_id)

edit_prompt = "The Colosseum is located in the city of"
base_prompt = "The Louvre is located in the city of"
batch = [edit_prompt,base_prompt]

edit_tok = 5
base_tok = 6

def hook_fn(module, input, output):
    output[0][1,base_tok,:] = output[0][0,edit_tok,:]

layer = lm.model.layers[5]
hook = layer.register_forward_hook(hook_fn)
inputs = tokenizer([edit_prompt,base_prompt], return_tensors="pt", padding=True)
output = lm(**inputs)
hook.remove()

last = output["logits"][-1,-1].argmax()
prediction = tokenizer.decode(last)
print(prediction)
```

Code Example 2: An intervention implemented with standard PyTorch hooks. Lines 1-4 load a language model from huggingface, and lines 7-9 define two prompts. In lines 11-12, we note the token indices for the last token of the subjects in each prompt. Lines 14-15 define a PyTorch hook that replaces the hidden state in the base prompt with the hidden state from the edit prompt using the respective last-subject-token indices. The experiment is carried out in lines 17-21, and we collect and print the output of the model in lines 23-25.

```python
from nnsight import LanguageModel
model_id = "meta-llama/Meta-Llama-3.1-8B"
lm = LanguageModel(model_id)

edit_prompt = "The Colosseum is located in the city of"
base_prompt = "The Louvre is located in the city of"
batch = [edit_prompt,base_prompt]

edit_tok = 5
base_tok = 6

layer = lm.model.layers[5]
with lm.trace(batch) as tracer:
    layer.output[0][1,base_tok,:] = layer.output[0][0,edit_tok,:]
    output = lm.output.save()

last = output["logits"][-1,-1].argmax()
prediction = lm.tokenizer.decode(last)
print(prediction)
```

Code Example 3: An intervention implemented with the NNsight API. This code defines the same intervention as in Code Example 2. Lines 1-3 load a language model from huggingface, and lines 5-7 define two prompts. In lines 9-10, we note the token indices for the last token of the subjects in each prompt. The experiment is carried out in lines 12-15, where we replace the hidden state in the base prompt with the hidden state from the edit prompt, using the respective last-subject-token indices. We collect and print the output of the model in lines 17-19.

Code Example 4 shows an NNsight implementation for attribution patching (Kramár et al., 2024), a method that involves both causal interventions and computation of gradients within the model.

```
1  def get_batch_states_and_grads(model, submodules, submod_names, batch, batch_target_ids):
2      hidden_states_cache = {}
3      grads_cache = {}

5      with model.trace(batch, remote=True) as tracer:
6          for submodule in submodules:
7              if 'neurons' in submod_names[submodule]:
8                  x = submodule.input
9              else:
10                 x = submodule.output[0]
11             hidden_states_cache[submodule] = x.save()
12             grads_cache[submodule] = x.grad.save()
13         neg_log_likelihood = metric_nll(model, batch_target_ids).save()
14         neg_log_likelihood.sum().backward()

16     hidden_states = {k : v.value[:,token_idx] for k, v in hidden_states_cache.items()}
17     grads = {k : v.value[:,token_idx]  for k, v in grads_cache.items()}
18     return hidden_states, grads, neg_log_likelihood
```

Code Example 4: An NNsight implementation for attribution patching (Kramár et al., 2024)

In Code Example 5, we show an example of how to implement and train a LoRA (Hu et al., 2021) adapter using a session context with remote execution via NDIF. Code Examples 6 and 7 demonstrate interventions on the gradients of model computations. And in Code Example 8, we show an what training a linear probe remotely via NDIF might look like to predict output of layer 1 using the output of layer 0.

```
1  import torch

3  from nnsight.envoy import Envoy
4  from torch.utils.data import DataLoader

6  # We will define a LORA class.
7  # The LORA class call method operations are simply traced like you would normally do in a .
       trace.
8  class LORA(nn.Module):
9      def __init__(self, module: Envoy, dim: int, r: int) -> None:
10         """Init.

12         Args:
13             module (Envoy): Which model Module we are adding the LORA to.
14             dim (int): Dimension of the layer we are adding to (This could potentially be
        auto populated if the user scanned first so we know the shape)
15             r (int): Inner dimension of the LORA
16         """
17         super(LORA, self).__init__()
18         self.r = r
19         self.module = module
20         self.WA = torch.nn.Parameter(torch.randn(dim, self.r), requires_grad=True).save()
21         self.WB = torch.nn.Parameter(torch.zeros(self.r, dim), requires_grad=True).save()

23     # The Call method defines how to actually apply the LORA.
24     def __call__(self, alpha: float = 1.0):
25         """Call.

27         Args:
28             alpha (float, optional): How much to apply the LORA. Can be altered after
        training for inference. Defaults to 1.0.
29         """

31         # We apply WA to the first positional arg (the hidden states)
32         A_x = torch.matmul(self.module.input[0][0], self.WA)
33         BA_x = torch.matmul(A_x, self.WB)

35         # LORA is additive
36         h = BA_x + self.module.output

38         # Replace the output with our new one * alpha
39         # Could also have been self.module.output[:] = h * alpha, for in-place
40         self.module.output = h * alpha

42     def parameters(self):
43         # Some way to get all the parameters.
44         return [self.WA, self.WB]

46  # get the model
47  llama = LanguageModel("meta-llama/Meta-Llama-3.1-70B")

49  # We need the token id of the correct answer.
```

```
50  answer = " Paris"
51  answer_token = llama.tokenizer.encode(answer)[1]
52  # Inner LORA dimension
53  lora_dim = 4

55  # Module to train LORA on
56  module = llama.model.layers[-1].mlp

60  # The LORA object itself isn't transmitted to the server. Only the forward / call method.
61  # The parameters are created remotely and never sent only retrieved
62  with llama.session(remote=True) as session:

64      # Create dataset of 100 pairs of a blank prompt and the " Paris " id
65      dataset = [["_", answer_token]] * 100

67      # Create a dataloader from it.
68      dataloader = DataLoader(dataset, batch_size=10)

70      # Create our LORA on the last mlp
71      lora = LORA(module, dim, lora_dim)

73      # Create an optimizer. Use the parameters from LORA
74      optimizer = torch.optim.AdamW(lora.parameters(), lr=3)

76      # Iterate over dataloader using .iter.
77      with session.iter(dataloader, return_context=True) as (batch, iterator):

79          prompt = batch[0]
80          correct_token = batch[1]

82          # Run .trace with prompt
83          with llama.trace(prompt) as tracer:

85              # Apply LORA to intervention graph just by calling it with .trace
86              lora()

88              # Get logits
89              logits = llama.lm_head.output

91              # Do cross entropy on last predicted token and correct_token
92              loss = torch.nn.functional.cross_entropy(logits[:, -1], batch[1])
93              # Call backward
94              loss.backward()

96          # Call methods on optimizer. Graphs that arent from .trace (so in this case session
        and iterator both have their own graph) are executed sequentially.
97          # The Graph of Iterator here will be:
98          # 1.) Index batch at 0 for prompt
99          # 2.) Index batch at 1 for correct_token
100         # 3.) Execute the .trace using the prompt
101         # 4.) Call .step() on optimizer
102         optimizer.step()
103         # 5.) Call .zero_grad() in optimizer
104         optimizer.zero_grad()
105         # 6.) Print out the lora WA weights to show they are indeed changing
106         iterator.log(lora.WA)
```

Code Example 5: Using NNsight to train a LORA with remote execution, the Session context, and iterative interventions. NNsight supports creating parameters and optimizers remotely. Using 'model.trace' as a forward pass, a dataset can be iterated over to optimize the remote parameters. The normal workflow of calculating a loss, executing a backwards pass, and taking optimizations steps is expressed just as if this were code executing on the user's local workstation. After executing the training loop, the optimized parameters are sent back to the user's workstation to be used locally, or to be sent back to the remote server for future applications of parameters.

```
1   import nnsight
2   from nnsight import NNsight
3   from collections import OrderedDict
4   import torch

6   input_size = 5
7   hidden_dims = 10
8   output_size = 2

10  net = torch.nn.Sequential(
11      OrderedDict(
```

```
12          [
13              ("layer1", torch.nn.Linear(input_size, hidden_dims)),
14              ("layer2", torch.nn.Linear(hidden_dims, output_size)),
15          ]
16      )
17  ).requires_grad_(False)
18  tiny_model = NNsight(net)
19  input = torch.rand((1, input_size))

21  with tiny_model.trace(input):

23      # We need to explicitly have the tensor require grad
24      # as the model we defined earlier turned off requiring grad.
25      tiny_model.layer1.output.requires_grad = True

27      # We call .grad on a tensor Proxy to communicate we want to store its gradient.
28      # We need to call .save() since .grad is its own Proxy.
29      layer1_output_grad = tiny_model.layer1.output.grad.save()
30      layer2_output_grad = tiny_model.layer2.output.grad.save()

32      # Need a loss to propagate through the later modules in order to have a grad.
33      loss = tiny_model.output.sum()
34      loss.backward()

36  print("Layer 1 output gradient:", layer1_output_grad)
37  print("Layer 2 output gradient:", layer2_output_grad)
```

Code Example 6: Applying backpropagation and accessing gradients with respect to a loss.

```
1  import nnsight
2  from nnsight import NNsight
3  from collections import OrderedDict
4  import torch

6  input_size = 5
7  hidden_dims = 10
8  output_size = 2

10  net = torch.nn.Sequential(
11      OrderedDict(
12          [
13              ("layer1", torch.nn.Linear(input_size, hidden_dims)),
14              ("layer2", torch.nn.Linear(hidden_dims, output_size)),
15          ]
16      )
17  ).requires_grad_(False)
18  tiny_model = NNsight(net)
19  input = torch.rand((1, input_size))

21  with tiny_model.trace(input):

23      # We need to explicitly have the tensor require grad
24      # as the model we defined earlier turned off requiring grad.
25      tiny_model.layer1.output.requires_grad = True

27      tiny_model.layer1.output.grad[:] = 0
28      tiny_model.layer2.output.grad = tiny_model.layer2.output.grad * 2

30      layer1_output_grad = tiny_model.layer1.output.grad.save()
31      layer2_output_grad = tiny_model.layer2.output.grad.save()

33      # Need a loss to propagate through the later modules in order to have a grad.
34      loss = tiny_model.output.sum()
35      loss.backward()

37  print("Layer 1 output gradient:", layer1_output_grad)
38  print("Layer 2 output gradient:", layer2_output_grad)
```

Code Example 7: Zeroing out the grad of one layer, and doubling the grad of a second layer.

```
1  import torch
2  from torch.utils.data import DataLoader
3  from nnsight import LanguageModel, list, log

5  model = LanguageModel("meta-llama/Meta-llama-3.1-405B")

7  with model.session(remote=True) as session:
8      features = 4096
9      probe = torch.nn.Linear(features, features).save()
```

```
10      dataset = DataLoader(
11          ["some", "text", "to", "train", "on"], batch_size=2, shuffle=True
12      )
13      optimizer = torch.optim.Adam(probe.parameters(), lr=0.003)

15      for epoch in list(range(50)):
16          for batch in dataset:
17              with model.trace(batch) as tracer:
18                  input = model.model.layers[0].output[0]
19                  probe = probe.to(input.device)
20                  prediction = probe(input)
21                  output = model.model.layers[1].output[0]
22                  loss = torch.nn.functional.mse_loss(prediction, output)
23                  loss.backward()

25              optimizer.step()
26              optimizer.zero_grad()

28          log(loss)
29          log(probe.weight)
```

Code Example 8: Sample code showing the remote training of a linear probe using NDIF.

# D  PERFORMANCE

In this section, we report additional results evaluating the performance of NNsight and NDIF in various settings. These include setup and runtime across different model sizes of OPT (Zhang et al., 2022), comparing the setup and runtime of HPC and NDIF for Llama-3.1 models (Dubey et al., 2024), and the response time for NDIF requests as a function of the number of users.

## D.1  HPC VS. NDIF

Table 2: Setup-time and runtime comparison of HPC and NDIF across OPT parameter sizes

| **OPT** | Parameters | Setup Time | Runtime |
|---|---|---|---|
| HPC | 125m | $0.635 \pm 0.035$ | $0.021 \pm 0.001$ |
| | 350m | $0.847 \pm 0.058$ | $0.041 \pm 0.001$ |
| | 1.3b | $1.242 \pm 0.016$ | $0.072 \pm 0.004$ |
| | 2.7b | $1.866 \pm 0.003$ | $0.123 \pm 0.011$ |
| | 6.7b | $3.474 \pm 0.023$ | $0.257 \pm 0.008$ |
| | 13b | $5.971 \pm 0.021$ | $0.429 \pm 0.018$ |
| | 30b | $11.338 \pm 0.073$ | $0.913 \pm 0.016$ |
| | 66b | $23.697 \pm 0.079$ | $1.889 \pm 0.040$ |
| NDIF | 125m | $0.149 \pm 0.008$ | $0.541 \pm 0.123$ |
| | 350m | $0.304 \pm 0.015$ | $0.588 \pm 0.178$ |
| | 1.3b | $0.409 \pm 0.213$ | $0.627 \pm 0.099$ |
| | 2.7b | $0.358 \pm 0.054$ | $0.494 \pm 0.085$ |
| | 6.7b | $0.317 \pm 0.011$ | $0.550 \pm 0.080$ |
| | 13b | $0.370 \pm 0.026$ | $0.773 \pm 0.136$ |
| | 30b | $0.343 \pm 0.013$ | $1.295 \pm 0.104$ |
| | 66b | $0.423 \pm 0.095$ | $2.739 \pm 0.154$ |

Table 3: Runtime comparison of NNsight using execution on an HPC and remote execution via NDIF.

| Framework | **Activation Patching** | |
|---|---|---|
| | Llama-3.1-8B | Llama-3.1-70B |
| `NNsight (HPC)` | $0.333 \pm 0.013$ | $1.992 \pm 0.056$ |
| `NNsight (NDIF)` | $0.998 \pm 0.134$ | $2.369 \pm 0.117$ |

Table 4: Time in seconds to load the model into memory.

| Framework | **Loading Weights** | |
|---|---|---|
| | Llama-3.1-8B | Llama-3.1-70B |
| `NNsight (HPC)` | $5.991 \pm 0.192$ | $43.617 \pm 1.118$ |
| `NNsight (NDIF)` | $0.518 \pm 0.005$ | $0.695 \pm 0.056$ |

## D.2  LOAD-TESTING NDIF FOR CONCURRENT USERS

In this section, we assess the performance and scalability of NDIF with an increasing number of concurrent users (and thereby requests). We use Llama-3.1-8B (Dubey et al., 2024) as the model being served on NDIF for this evaluation, which was hosted on a 48 GB RTX 6000 Ada GPU.

To simulate a single user, we construct a NNsight request with a prompt containing up to 24 tokens that accesses and saves the output of a layer of Llama-3.1-8B selected uniformly at random. Code

Example 9 shows how we generate sample prompts and requests for a single user, saving the response time of the request sent to and returned by NDIF.

```python
def layer_selector(self) -> None:
    """Simulate selecting activations of an intermediate layer."""
    start_time = time.time()
    try:
        layer : int = random.randint(0, self.n_layers - 1)
        query : str = 'hello ' * random.randint(1, 24)
        with self.model.trace(query, remote=True):
            output = self.model.model.layers[layer].output.save()
    finally:
        elapsed_time = time.time() - start_time
        # Append to a list visible locally to this process
        self.user.environment.request_times.append(elapsed_time)
```

Code Example 9: Code to generate NDIF requests and measure the response time for many users.

Then, for $N \in \{1, ..., 100\}$, we simulate $N$ concurrent users submitting requests to the NDIF server, all running as separate processes, and record the response time for each user's request.

The results can be seen in Figure 9, where we plot the response time of the NDIF system as a function of the number of concurrent users. We find that as the number of users increases, the increase in the median response time (shown in black) is approximately linear. Furthermore, as more concurrent users access the system, variance in response time increases.

This experiment was done using an implementation of the system that creates a queue for each subsequent user, and runs multiple forward passes (one for each user). There are future plans to allow true parallel execution of user requests in the same batch.

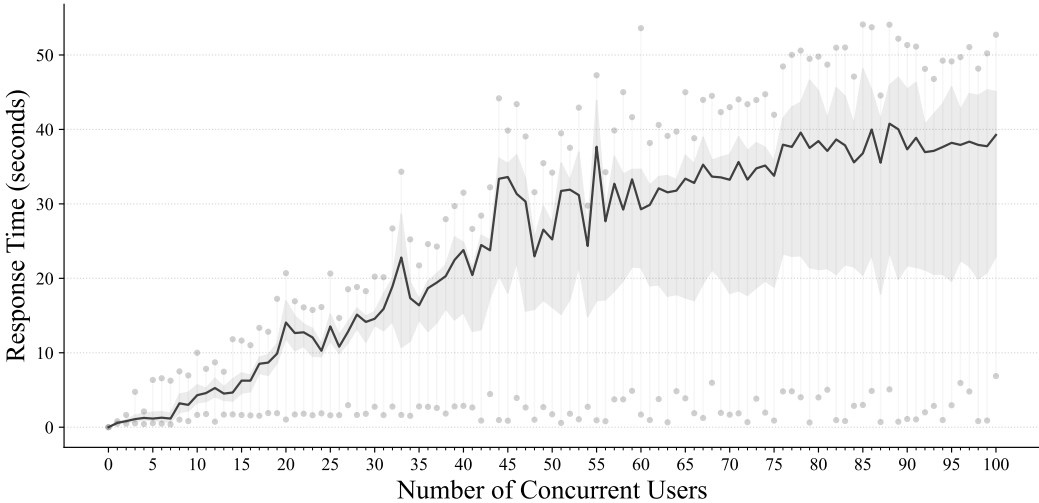

Figure 9: Speed and robustness of NDIF response time to changes in concurrent user request count. Tests simulated $N = 1$ to 100 concurrent users on Llama-3.1-8B requesting outputs from uniformly random intermediate layers over a 30-second window. The gray area shows 25% and 75% quantiles. Points show the minimum and maximum times for each user count. Line plot shows median response time.

# E  EQUIVALENCE OF THEANO'S MANY-TO-MANY AND NNSIGHT'S MANY-TO-ONE APPLY NODES

We demonstrate that Al-Rfou et al. (2016)'s many-to-many computation graph structure—where an apply node can output multiple variable nodes—is computationally equivalent to a graph in which each apply node outputs only a single variable node. This equivalence holds by considering the many-to-many structure as a special case of a many-to-one structure, where the single output is a tuple (or concatenation) of variable nodes.

Let $a^{(\text{in})}$ denote an apply node in the many-to-many setting, and let $v_1, \ldots, v_m$ represent the $m$ variable node outputs of $a^{(\text{in})}$. Suppose these variable nodes serve as inputs to $n$ downstream apply nodes $a_1^{(\text{out})}, \ldots, a_n^{(\text{out})}$ in the computation graph.

We now construct a new graph that adheres to the many-to-one structure, where each apply node has exactly one output. Let $b^{(\text{in})}$, $v$ , and $b_1^{(\text{out})}, \ldots, b_n^{(\text{out})}$ denote corresponding nodes in the transformed graph, defined as follows:

1. The apply node $b^{(\text{in})}$ outputs a single variable node $v$, which is defined as the concatenation (or tuple) of the original outputs: $v = (v_1, \ldots, v_m)$.

2. Each downstream apply node $b_i^{(\text{out})}$ takes as input $v$ and any other variables that were inputs to $a_i^{(\text{out})}$ in the original graph. The node $b_i^{(\text{out})}$ performs the same computation as $a_i^{(\text{out})}$, but accesses the necessary components of $v$ corresponding to its original inputs. Specifically, if $a_i^{(\text{out})}$ originally took inputs $v_{i_1}, \ldots, v_{i_k}$, then $a_i^{(\text{out})}$ extracts $v_{i_1}, \ldots, v_{i_k}$ from $v$ and processes them accordingly.

This process generalizes inductively to any depth of the computation graph. In the event that one of the output variable nodes $v_i$ needs to be used elsewhere in the computation graph (e.g., as input to a node that does not directly accept $v$), the transformation can be handled by introducing an additional apply node, $b^{(\text{extract})}$. This node takes $v$ as input and outputs $v_i$, ensuring that the structure of the original computation graph is maintained.

Figure 10 shows an example of how getters and setters interact with an intervention graphs variable and apply nodes on an abstract computation graph.

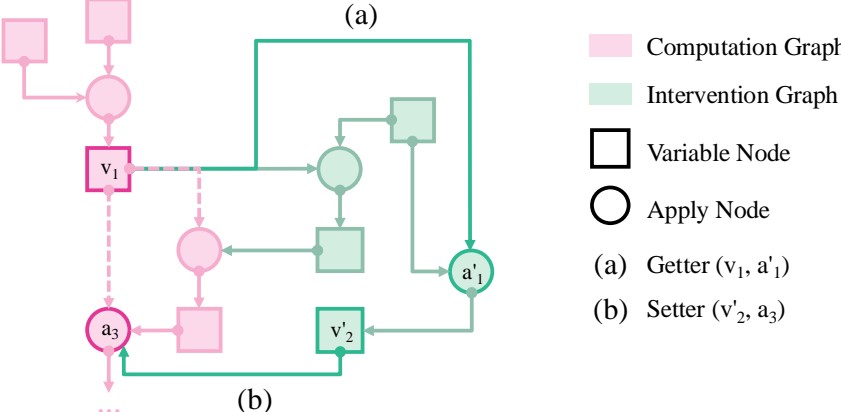

Figure 10: Augmenting the computation graph $C$ (pink graph on the left) with an intervention graph (green, right). (a) A getter edge brings the output of a variable node $v_1$ from the computation graph to an apply node $a_1'$ in the intervention graph. (b) A setter edge sends the output of $v_2'$ in the intervention graph back to $a_3$ in the computation graph.

