# OpenReview forum: "NNsight and NDIF: Democratizing Access to Open-Weight Foundation Model Internals"
_ICLR.cc/2025/Conference — ICLR 2025 Poster_

### Official Review · Reviewer_JASr · 2024-10-31

**Soundness:** 3
**Presentation:** 3
**Contribution:** 4
**Rating:** 8
**Confidence:** 5

**Summary:**

The paper introduces NNsight and NDIF, a framework and a cloud inference service designed to facilitate research on large-scale AI models, and enable study of the representations and computations learned by large neural networks. This work proposes a method for organizing experiments on very large models, reducing engineering burden, enhancing reproducibility, and enabling low-cost communication with remote models. The NNsight is an open-source implementation of the intervention graph architecture that extends PyTorch to support transparent model interventions without requiring local storage or management of model parameters. And the NDIF is an open-source cloud inference service that supports the NNsight by providing behind-the-scenes user sharing of model instances to reduce the costs of large-scale AI research.

**Strengths:**

1. The intervention graph architecture and the NNsight system provide a practical and efficient way to decouple experimental and engineering code, making it easier to conduct complex experiments on studying the intermediate representations and gradients of large models.
2. The paper includes a thorough performance evaluation comparing NNsight and NDIF with HPC and Petals, demonstrating the efficiency and effectiveness of the proposed solutions.
3. The performance comparisons between NNsight and other frameworks like baukit, pyvene, TransformerLens show the similar efficiency of NNsight.
4. The remote execution of NNIF provides the easy usage for users.

**Weaknesses:**

1. While the developed tools of NNsight and NDIF are very useful in scientific study of AI models, there is no founded insights of this work. I'm not sure whether this contribution is enough for the acceptance.
2. Exploiting the DAG representation is not novel. As indicated by the authors, [1] has proposed the DAG concepts and use it to accelerate computation. Besides, the Torch Profiler [2] and torch.jit.trace [3] can help to dissect the model. I do not see any obvious technical challenges to implement the NNsight. Could you elaborate more on such challenges and illustrate how you address them? And how the NNsight is different from the Torch Profiler and torch.jit.trace?
3. For the practical usage like shown in Figure 4, I do not see obvious advantages of NNsight than using hooks of Torch. The required number of code lines of them are similar. Could you also elaborate more about this?

[1] Theano: A python framework for fast computation of mathematical expressions. 2016
[2] PyTorch Profiler. https://pytorch.org/tutorials/recipes/recipes/profiler_recipe.html.
[3] Torch.jit.trace. https://pytorch.org/docs/stable/generated/torch.jit.trace.html.

**Questions:**

Please refer to the weaknesses and provide more clarifications.

---

> ### Author Response · Authors · 2024-11-19
> **To Reviewer JASr**
>
> Thank you for your review. We address your questions below.
>
> ### Foundational insights
> Our research is submitted in the “infrastructure, software libraries, hardware” subject area, because its intended contribution is focused on innovations in infrastructure and software that will enable future researchers to create foundational insights in machine learning. Our goal is to create the infrastructure that is necessary to enable other research; for example, at least four other submissions to ICLR this year use the NDIF infrastructure for research that would not have been possible without it.
>
> ### Novelty of DAG representation
> DAG representations of code predate Theano and other deep learning libraries. Whereas the innovation of previous deep learning libraries was to apply DAG to the implementation of automatic gradient and backpropagation, the innovation of NDIF is to apply DAG representations to the problem of remote experimentation on the internals of very large models.
>
> ### Comparing NDIF to the torch profiler and jit.trace
> These tools are focused on single-machine execution without providing a mechanism for altering a shared execution graph. The problem tackled by NNsight is to create graphs that express modifications of a larger DAG (appropriate for remote execution), which is a new problem setting that is not addressed by any previous tools.
>
> ### Advantages of NNsight over PyTorch hooks
> The main advantage of NNsight over pytorch hooks is that hooks do not allow for safe remoting or co-tenancy, because a hook would encourage researchers to create arbitrary code execution (up to and including erasing files or rebooting the machine) that cannot safely be done on shared compute. NNsight creates a safely remotable graph-creation API that is nevertheless just as succinct and nearly as fully expressive as hooks.  See code examples  4, 5, 6, 7, and 8 in appendix C for more complex examples. These show a number of situations in which using the NNsight API is powerful enough to express complex interventions on neural network internals. By enabling simple remoting, NNsight removes the burden of dealing with multi-device parallelization, sharding, quantization, and other engineering concerns when dealing with huge models.

---

> > ### Comment · Reviewer_JASr · 2024-11-25
> > **Thanks for the responses**
> >
> > I agree that the DAG representation is used for automatic gradient and backpropagation in other DL libraries. However, utilizing this technique in the NNIF is trivial and could not be seen as a major contribution.
> > I agree that the NNIF and NNsight would be helpful for the LLM interpretation research. However, the system design and its implementation might not be enough to support its contribution. From a system perspective, maybe we need to see its real-world applications and the real-world performance data (like latency, throughputs and others) in details.
> >
> > Could you elaborate more on providing 'a mechanism for altering a shared execution graph'? I think this part is interesting and it would be better to be emphasized more. How the users' applications need a shared execution graph? I guess that users may need some probes to study the features and gradients, like using linear probes. Thus, the execution graphs are changed. Could you show more examples and answer this?

---

> ### Author Response · Authors · 2024-11-26
> **Response**
>
> We appreciate the reviewer highlighting the need to clarify our technical contribution. The core innovation of our work lies not in the use of DAGs, but in solving a fundamentally new problem: How can multiple researchers safely and efficiently share access to a single preloaded large model instance, in settings where they may need to be able to access and modify model internals?
>
> Previous solutions like PyTorch hooks assume single-user access to dedicated hardware. This assumption breaks down for modern large language models, where dedicating an entire GPU cluster to each researcher is economically infeasible. For example, loading a 70B parameter model takes 43 seconds and requires multiple GPUs with sufficient memory for weights, activations, and gradients (Table 4). With hooks, each researcher needs their own copy of the model, their own GPU allocation, and must handle sharding and device management themselves.
>
> We solve this by turning experiment modifications into serializable, storable "diffs" that can be safely interleaved with a single shared model instance. This is separate from torch.fx (and torch Dynamo, and similar systems) in that those systems are primarily concerned with converting a model’s forward method into a DAG, without any modifications. For example, one unique challenge for NDIF is the interrupted nature of execution for a user’s intervention code: Interventions are not always executed in one pass sequentially, but in pieces depending on when an intervention’s corresponding module is executed. Even more challenging is that modules can be executed an arbitrary number of times, and so proper care needs to be taken to execute interventions properly. Our approach allows higher-level operations such as running forward passes on models, looping over a dataset, creating and applying an optimizer, and more.
>
> The resulting system enables multiple researchers to run different experiments concurrently on the same hardware without interference. The results are substantial: total execution time for OPT-66B drops from 25.6s to 3.2s (Table 2) by eliminating repeated weight loading and device management overhead.
>
> Making this system work required solving several technical challenges that don't exist in the single-user setting:
> 1. Ensuring safe multi-user co-tenancy by preventing arbitrary code execution (which PyTorch hooks allow)
> 2. Managing complex dependencies when multiple intervention graphs interact with the same model
> 3. Implementing efficient request batching across users
> 4. Maintaining differentiability while keeping shared memory usage low
>
> To elaborate on mechanisms for altering a shared execution graph, we can track which "batch groups" - specific slices of the model's input and output - correspond to each user's interactions. This allows us to isolate and apply each user's interventions, like gradient manipulation, only to the relevant parts of the shared model execution. Future implementations will allow this for multiple experiments on the same forward pass.
>
> Our code examples in Appendix C also demonstrate some examples of altering shared execution graphs. For instance, Example 6 shows LoRA fine-tuning running remotely on a shared model, while Examples 7-8 show how multiple users can safely manipulate gradients without interfering with each other. Code Example 9 shows an example of a linear probe training experiment expressed via NNsight and executable via NDIF.
>
> On real-world applications - we believe the scheduling problem we solve is critical for democratizing access to large model research, as evidenced by the four other ICLR submissions this year that rely on our infrastructure. We also point you to our response to reviewer PJHU, who asked a similar question.

---

> > ### Comment · Reviewer_JASr · 2024-11-27
> > **Thanks for your clarification**
> >
> > Very comprehensive clarification.
> >
> > I have following suggestions for your revision:
> > - 1. Highlighting the challenge on the user-side coding: `With hooks, each researcher needs their own copy of the model, their own GPU allocation, and must handle sharding and device management themselves.` Especially, addressing the problem of `how to implement the sharding and device management themselves` can be a main technical contribution which is not easily supported by the PyTorch Hook, DeepSpeed or Megatron.
> > - 2. Highlighting the challenge: `Interventions are not always executed in one pass sequentially, but in pieces depending on when an intervention’s corresponding module is executed. `. There might be some subtle details of the user intervention that cannot be simply implemented using the PyTorch Hook. This should be illustrated with much more details.
> > - 3. Highlighting the `mechanisms for altering a shared execution graph`.
> >
> > And could you please elaborate the above points more? I think it would be better to add a system design section to show how they are implemented. And there can be a system design figure to show how different system modules depend on others. Specifically, how is the computing graph defined and represented? How does the automatic `device management` (without users' management) is implemented built upon the DAG? This might be a most challenging part, how do you implement the communication between different devices, how to optimize the throughtput and latency under this scenario? What is the low-level framework dependency? Only Torch? or including DeepSpeed or Megatron and others?
> >
> > I would like to increase my score with your new illustration.

---

> ### Author Response · Authors · 2024-11-28
> **New updates**
>
> Thank you for your continued feedback. We have uploaded a revised version of our manuscript. We renamed Appendix B to “System Design”. The original subsection on the design of NNsight that used to be called “Implementation Details” is now “NNsight Design”. We have added a new subsection on NDIF design. Furthermore, we have added a new Figure 8 that shows how the intervention graph is executed by NDIF, and how the remote system works with respect to sharding and memory.
>
> The NNsight System Design subsection newly clarifies how PyTorch modules are wrapped to create the intervention graph. It also explains how the intervention graph is implemented, convenience methods for particular types of neural networks, how PyTorch modules are wrapped, how values are saved to the user, how debugging works, and how multiple forward pass requests can be batched.
>
> The new NDIF System Design subsection explains the implementation details you requested around processing and graph distribution, distributed memory management, how distributed interventions are executed, and resource management with co-tenancy.
>
> The new Figure 8 gives an overview of the implementation details for the whole system, and show how we use our HTTP frontend to pass intervention graphs and parallelize on the backend.
>
> We respond to your questions directly in the next comment due to character limits.

---

> ### Author Response · Authors · 2024-11-28
> **Questions**
>
> We also answer your questions directly below.
>
> **How is the computing graph defined and represented?**
> The Intervention Graph paragraph in Appendix B gives insight on the definition and representation of the intervention graph in practice, and Section 3.1 gives a mathematical definition.
>
> The intervention graph, in practice, is defined by a set of NNsight Node objects. A Node represents a single operation to be executed. Nodes have unique identifiers and arbitrary arguments, which may contain other Nodes as dependencies. During interleaving, when a Node is executed, it decrements its remaining dependencies and listeners. When a Node's dependencies reach zero, it executes immediately. Nodes are freed when they have no remaining listeners, optimizing memory usage, unless the node is configured to store its value.
>
> The graph is built through Proxy objects that manage operations on Nodes by overriding Python and PyTorch magic methods. Each operation creates and adds a new Node to the graph, returning the corresponding Proxy. The root intervention Nodes created with Envoys are the only Nodes executed outside this dependency chain.
>
> More details on this setup can be found in Appendix B.
>
> **Communicating the intervention graph between local and NDIF**
>
> When a user creates an intervention, NNsight constructs the intervention graph with the methods above and described in Appendix B. This graph is serialized into a custom JSON format. We then send the graph to NDIF with an HTTP server front-end. When NDIF receives a graph, it interfaces with Ray, a distributed computing infrastructure. The request (and intervention graph) is put on a queue for Ray to process. When the graph reaches the head of the queue, it is sent to the correct shard deployment, where the graph is deserialized and interleaved with the preloaded model's computation.
>
> The interleaving process works as follows: First, the head node receives the serialized intervention graph from NNsight. The intervention is distributed across worker nodes based on the tensor parallelism configuration. Then, each worker executes its portion of the model while interleaving the intervention operations at the appropriate points. Interleaving happens by, for each module execution point, checking if there are any intervention nodes that depend on the module. If yes, those module’s nodes are executed when their dependencies are satisfied. The special save method in NNsight marks the results that should be sent back to the user. Then normal module execution continues. Once execution completes, all outputs marked with save are aggregated back through the head node and sent back to the user.
>
> We use this approach to minimize data transfer overhead compared to traditional approaches that would require sending full tensors or activation states between client and server.
>
> For more on how interventions work on a sharded model, see Request Processing and Graph Distribution in the updated Appendix B.2
>
> **How does automatic device management work on NDIF?**
>
> As users do not have custody of model parameters, they cannot interact with tensor device management on the NDIF service. Instead, NDIF handles automatic device management in two steps:
>
> As shown in step 4 of figure 8 in our revised manuscript, the head node (Shard 0) sends the serialized intervention graph to each model shard, where it is then deserialized. During deserialization, each shard annotates the created tensors with its default local device ID, the singular GPU a shard has custody over. Therefore all tensors created are automatically annotated with the shard’s device ID it is processed on.
> Before any operation is executed within the intervention graph, all arguments (positional and keyword) are searched for Tensors. We note the set of all devices amongst the Tensors found, and if there are more than one device found, we move the Tensors onto one of the devices.
>
> For more on tensors management in a distributed setting please see Distributed Model Management  in the updated Appendix B.2
>
>
> **What are the dependencies of our framework?**
>
> NNsight is primarily built on Torch. Other dependencies include accelerate, transformers, diffusers, and torchvision, as well as ease-of-use packages like Pydantic. All of NNsight’s dependencies are simple Python, and users interact with NDIF only through NNsight. To interact with the system, users therefore only need to run NNsight with a pip install.
>
> NDIF primarily depends on NNsight, torch distributed, and Ray (https://github.com/ray-project/ray) - specifically Ray Serve. We are not using DeepSpeed or Megatron. NDIF uses NCCL as the backend for torch distributed. Ray’s distributed computing infrastructure is used for coordination. We have automatic cleanup procedures in place for memory management. There are other minor dependencies with varying levels of importance, such as an object store (currently redis) and a fastapi backend.

---

> > ### Comment · Reviewer_JASr · 2024-11-28
> > **Thanks for the new updates**
> >
> > I appreciate for the new revision and the responses. Very nice and detailed explanation. I'd like to increase my score. Now I can understand the detailed system design. I find that the core design of the engine behind the NNIF and NNsight achieve a similar goal of recent topic of decentralized training about how to distribute the operations into different clusters or remote devices [1, 2, 3]. To support such an automatic module allocation (or says distributed model management) and remote execution, there should be a high-level wrapper to manage each module (operation) and its dependency in the whole DAG (Section 3 in [1]). It looks that NNIF and NNsight implement this by the mentioned designs of `Envoy`, `Proxy` and `Protocol`. Therefore, the core engine of NNIF and NNsight can be exploited and extended to support decentralized training like [1, 2, 3]. Authors can add some discussion of this scenario and compare with the design of [1,2] as its broader impact.
> >
> > Besides, there are some follow-up suggestions and questions:
> > 1. **The system design section can be added into the main text (with a shorter version, considering the limited space)**;
> > 2. **How do you manage the communication between different devices?** Do you directly obtain results from Ray? There is a problem that the Ray is a stateless framework, it seems that the processes in Ray will not store features or gradients in processes. Based on this, how do you implement training? Because the training workers need to store the features for backward propagation, and store gradients for model updating. If I understand correctly, the different shards (1...N) in Figure 8 will not communicate with others, but only with shard 0? It seems that such a communication scheme is low efficient and cause communication congestion on the shard 0.
> > 3. In the model abstraction part, **can the current design directly support users to use the `transformers` library?** It seems that the design of NNsight cannot support seamlessly integrating `transformers` into NNsight. This may lower the easy usage. Future improvements can consider this problem.
> >
> > [1] FusionLLM: A Decentralized LLM Training System on Geo-distributed GPUs with Adaptive Compression. In Arxiv 2024.
> >
> > [2] FusionAI: Decentralized Training and Deploying LLMs with Massive Consumer-Level GPUs. In Arxiv 2023.
> >
> > [3] Petals: Collaborative inference and fine-tuning of large models. 2023.

---

> ### Author Response · Authors · 2024-11-28
> **Responses**
>
> Thank you for the feedback and for your kind reconsideration of the score.
>
> Our system is not primarily designed for training - it is designed to support the interpretability community in investigating model internals. For example, a common use case would be activation patching (Zhang & Nanda, 2024), where activations from one prompt are swapped with activations from another (see Code Examples 4 and 5 and others in Appendix C). However, we will plan on updating the system design section with some discussion on training based on your feedback. We will do this upon final release of the paper, since the PDF submission deadline has passed.
>
> On the transformers library: Yes, NNsight comes with convenience functions to use `transformers` models. Any HuggingFace model can be used immediately with NNsight by passing the model ID in as a string. See the "Model Abstractions" section of B.1, or, for instance, the code in Figure 4 where we load a model in from Huggingface for use with the system.
>
> The space in the main text is limited, and we would have to cut essential text to incorporate even a shortened system design section. We believe that the level of detail it requires is best suited for the appendix.

---

### Official Review · Reviewer_PJHU · 2024-11-03

**Soundness:** 3
**Presentation:** 3
**Contribution:** 2
**Rating:** 6
**Confidence:** 4

**Summary:**

This work presents a new PyTorch-compatible library for remotely accessing the internal structures of deep-neural-network-based models such as large language models (LLMs) and altering the way they operate. Using this library, it is possible to produce intervention code which is hooked into the original model to read or replace original model parameters and activations. This intervention code is translated into a directed acyclic intervention graph that augments the computational graph of the original model.  Others have proposed similar, but less flexible mechanisms. For instance, pyvene (Wu et al., 2024) supports dictionary-based intervention definitions instead of the code-based interventions proposed by this work, which can also be transformed into graph-based representations and further optimised, e.g., by TorchScript. Petals (Borzunov et al., 2023), on the other hand, does not support remote execution of the intervention code, requiring it to be executed locally, which limits virtualisation opportunities and leads to additional communication overheads.

**Strengths:**

- The proposed approach involves some new mechanisms that complement the related work.
- By enabling remote execution of the intervention code the authors have demonstrated a superior performance compared to Petals.

**Weaknesses:**

An interesting real-life application of the infrastructure built is missing. For instance, the authors could consider demonstrating their system on a large-scale interpretability study or model editing task that would be infeasible without their infrastructure.

**Questions:**

It looks like the main technical contribution of this work is the enablement of efficient/virtualised remote interventions on LLMs. Could the authors elaborate on why they believe this contribution is relevant and impactful for the ICLR community?

---

> ### Author Response · Authors · 2024-11-19
> **To Reviewer PJHU**
>
> Thank you for your review. We address your questions below.
>
> ### Real-Life Applications
> NDIF is currently running a pilot deployment providing research access to the Llama 3.1 405B model to researchers from 10 diverse research groups at a variety of institutions (institutions omitted to preserve anonymity). NDIF is one of the few practical ways to conduct invasive research on large models at this scale. The deployment, which has been running since August 2024, has enabled several conference paper submissions under review that would not have been possible without access to the NDIF infrastructure.
>
> ### Relevance and Impact
> Despite the release of open-weight models at the order of Llama 3.1 405B, there has been almost no research engagement at that scale so far due to lack of necessary research infrastructure; Section 2 of the paper, and in particular Figure 2, document this situation. Limiting research engagement to the small number of research groups with the financial and engineering infrastructure to support research on large models is undemocratic and unhealthy for scientific discovery. The NDIF infrastructure is the first step in ameliorating this problem.

---

> ### Author Response · Authors · 2024-11-30
> **Addressed Concerns**
>
> We have updated the paper to clarify how multiple concurrent user experiments are processed on NDIF, to include new discussions on related works, we have added a new figure 8 to appendix B as well as a new Appendix B.2 to clarify how the NDIF system is implemented, and a new stress-testing experiment to Appendix D.2 to show how the system speed changes as users make more concurrent requests. See the "Paper Changes" update to all reviewers for more information, as well as discussion with other reviewers.
>
> We address your concerns about an interesting real-life application of the infrastructure as well as the impact and relevance for ICLR in the rebuttal. Please let us know if we have addressed them adequately.

---

### Official Review · Reviewer_RzZK · 2024-11-04

**Soundness:** 2
**Presentation:** 3
**Contribution:** 3
**Rating:** 6
**Confidence:** 3

**Summary:**

This paper introduces NNsight and NDIF, two open systems that provide efficient, transparent access to the internals of large neural networks for research purposes. NNsight extends PyTorch to offer deferred remote execution of intervention graphs, while NDIF serves as a scalable inference service that executes these intervention requests, enabling resource sharing among users. The work addresses challenges like limited access to state-of-the-art models and the significant resource demands of large-scale AI research by sharing resources.

**Strengths:**

* The intervention graph extends the model computational graph and decouples experimental design from model runtime, reducing engineering complexity.
* NDIF effectively shares GPU resources among researchers (co-tenancy), reducing cost and enabling large-scale experiments
* This is a novel idea with great potential benefit to the research community.

**Weaknesses:**

* The evaluation section is limited to the end-to-end performance with little detail on the system optimizations. Specifically, more information is needed to assess the performance and scalability of this system using an increasing number of users.
* ~~While NDIF addresses large-scale experiments on open models, it does not cover closed, proprietary models hosted proprietarily.~~
* Lack of discussion with relevant co-serving systems like S-LoRA (Sheng et al, MLSys 2024) and dLoRA (Wu et al, OSDI 2024).

**Questions:**

Could you please share more evaluation results on the performance and scalability of the proposed systems?

---

> ### Author Response · Authors · 2024-11-19
> **To Reviewer RzZK**
>
> Thank you for your review, we’re glad you found our contributions novel and beneficial for the research community. We have updated the manuscript as discussed in the global rebuttal, and we also address your specific concerns below.
>
> ### Assessing the performance and scalability of NDIF
> We are updating the manuscript with an experiment that tests the robustness of NDIF as more users submit requests. The test is to simulate N users sending requests to NDIF. For each user, we pick a random layer and prompt, and send the representations in that layer for that prompt back to the user. We increase N and report performance. We will send another update with more details when this experiment is complete.
>
> ### Closed, proprietary models
> We refer to the global rebuttal for discussion on this point. We have expanded the discussion in Lines 526-533 in Section 6 on this topic. We have also updated the title of the paper to be “NNsight and NDIF: Democratizing Access to Open-Weight Foundation Model Internals” to make it more clear that NDIF does not currently provide access to closed, proprietary models.
>
> ### Lack of comparison with S-LoRA and dLoRA
> Thank you for the pointers to S-LoRA and dLoRA. We have updated our discussion in the related work section to include this work. While these systems are focused on efficient serving of LoRA adapters, they are limited in scope and do not enable the wider range of model access and interventions that NNsight and NDIF are designed to support.

---

> ### Author Response · Authors · 2024-11-30
> **Addressed Concerns**
>
> We have added a new Figure 8 to Appendix B, as well as a new "NDIF Design" Appendix B.2 to clarify how the NDIF system is implemented.
>
> To address your question about evaluation and system optimizations, we have added a new "Load-Testing NDIF for Concurrent Users" Appendix Section D.2.
>
> To address your question about relevant co-serving systems, we have updated the related work section to include a discussion on S-LoRA and DLoRA.
>
> See the "Paper Changes" update to all reviewers for more information, as well as discussion with other reviewers. Please let us know if we have addressed your concerns adequately.

---

> > ### Comment · Reviewer_RzZK · 2024-12-02
> >
> > I appreciate the authors for the responses. I have gone through the paper changes and the discussions with other reviewers, I think the updates meet the criteria for acceptance and decide to upgrade my score.
> >
> > As a suggestion to the over-length issue, I would like to suggest the authors to a) reduce the size of Fig 2 and Fig 3 (potentially by fitting them into one row), and b) consider moving the discussion of interface/implementation (Section 3.2) to the Appendix. By doing this, you can leave more room for the key ideas (particularly on the design and motivations for the design), which are important since this submission is about a practical ML system.
> >
> > Thanks.

---

> > > ### Author Response · Authors · 2024-12-02
> > > **Suggestions**
> > >
> > > We thank you for your reconsideration of the score, as well as your suggestions. We are currently working on reducing the size of the figures and adjusting the text. Although we are unable to update the PDF due to the deadline, we will include these updates in the camera-ready version.

---

### Official Review · Reviewer_RQ4u · 2024-11-04

**Soundness:** 3
**Presentation:** 3
**Contribution:** 3
**Rating:** 6
**Confidence:** 4

**Summary:**

The paper introduces two systems NNsight and NDIF that collectively aim to reduce the developer and hardware costs of analyzing and modifying the inference behavior of open-source models. NNsight is an instrumentation framework for PyTorch models, while NDIF is an inference service that enables deferred execution of NNsight instrumentations on a remote shared model deployment. The paper provides evaluation comparing to inference baselines of shared (Petal) and non-shared HPC deployments.

**Strengths:**

- This paper is focusing on an important problem since tools that enable introspection of model internals are extremely valuable for ML research and applications.
- Opting for Pytorch-native tools is a great design choice as that significantly reduces development and integration burden for practitioners.
- Enabling resource sharing via NDIF service is very useful to democratizing access to SOTA models.

**Weaknesses:**

- The paper seems to overclaim, particularly in the title and abstract, that the work applies to foundation models in general, when in reality it only applies to open-source models, since model internals knowledge is required to create interventions. The authors should more carefully scope the claims.
- The evaluation does not study co-tenancy scenarios where the NDIF is servicing multiple NNsight requests. This is a significant oversight considering that the resource sharing benefits of NDIF is a major contribution of the work. The authors should include results showing not just the performance implications of servicing the multiple NNsight requests but also validating the correctness of the co-tenancy features.

**Questions:**

1. Do users submit a pair of NNsight request and an input prompt? Or how is the input prompt for exercising an intervention generated?
2. When multiple NNsight requests are submitted, are all the interventions applied to a single model instance for inference, or is a separate model instance created for each request?
3. If a single model instance is instrumented with multiple NNsight requests, how does NDIF ensure that a request that modifies model parameters does not affect other requests?
4. How is KV-cache managed for multiple NNsight requests?

---

> ### Author Response · Authors · 2024-11-19
> **To Reviewer RQ4u**
>
> Thank you for your review, and we are happy to hear that you agree with us on the importance of the problem. We address your concerns below.
>
> ### Applicability to foundation models.
> We have updated our paper title to clarify that our research architecture is built specifically for open-weight models by adding “open-weight” before “foundation model internals”. Lines 526-533 in Section 6 also discuss this issue, encouraging model providers to open access to their weights. We address this in the global rebuttal. The goal is to be able to scale research to the size of open-weight foundation models like Llama 3.1 405B; we use the term “foundation” here to include open-weight models as well.
>
> ### Co-tenancy evaluation.
> Thank you for pointing out this weakness; we agree that careful evaluation of the robustness of NDIF in co-tenancy is a necessary inclusion. As discussed in the global rebuttal, we are currently running an experiment to address this issue and we will send another update when it is complete.
>
> ### How is the input prompt for exercising an intervention generated?
> Inputs to the model are submitted upon opening the `tracing` context in the API. So if you were to use a language model called llama, on remote, you would write:
> ```
> with llama.trace("prompt goes here", remote=True):
>     <intervention code here>
> ```
> The input type is flexible. If you were using a model that took tensors as inputs, for instance, the string above would be replaced with a `torch.Tensor` object.
>
> Users submit both an input and their custom interventions simultaneously. NNsight and NDIF allow users of language models to simply pass in a string, and it will be automatically tokenized.
>
> ### Are multiple intervention requests applied to a single model instance or is a separate model instance created?
> User requests are sent to a single shared model instance, which is preloaded onto the NDIF server. We have updated the paper to clarify that this is what we are calling co-tenancy: keeping a single shared model instance open minimizes performance difficulties involved in weight-loading and model startup. In the current implementation, requests are queued and executed sequentially, in a way that ensures experiments can’t affect each other. We have updated Figure 5 and the text in section 3.3 to make sure there is no ambiguity about this, and we can elaborate with implementation details if you need them.
>
> NDIF’s intervention graph approach does allow for experiment parallelization on the same inference pass, which we will add in the future. This will be done by creating multi-user batches, where different user requests are allocated to different batch indexes. Implementation details are again available upon request.
>
> ### If a single model instance is instrumented with multiple NNsight requests, how does NDIF ensure that a request that modifies model parameters does not affect other requests?
> Good question. The NNsight API does not provide an interface for directly altering model parameters on NDIF. Instead, modifications to the model architecture and weights are simulated by augmenting the model’s original computation graph as a result of the interleaving produced by the user’s intervention graph. For instance, if you wish to ablate the activations of a layer, instead of setting the layer’s weights to 0, the augmented computation graph will pass the output to a user-defined node (on the intervention graph) responsible for ablating the activations and then pass it’s output as the input of the next layer. Parameter changes can similarly be simulated by adding other computations. The NDIF server maintains a whitelist of functions to prevent running non-permitted operations. This approach safeguards the co-tenancy principle of the NDIF service and ensures the correctness of all user intervention requests without interference. See lines 367-372, “Safe Co-tenancy”.
>
> ### How is KV-cache managed for multiple NNsight requests?
> The NNsight API includes a wrapper around the Huggingface Transformers AutoModelForCausalLM. Inside the tracing context, a generate method can be used to calculate multiple tokens in a single trace; this generate method uses the Transformers implementation of kv-caching under-the-hood. Entire user requests (many generated tokens) are considered one session and the KV cache is managed by transformers during the period where that user’s request is being processed.

---

> ### Author Response · Authors · 2024-11-30
> **Addressed Concerns**
>
> We have added a new Figure 8 to Appendix B, as well as a new "NDIF Design" to Appendix B.2 to clarify how the NDIF system is implemented.
>
> We have also added a new "Load-Testing NDIF for Concurrent Users" Appendix section D.2 to address your question about co-tenancy scenarios.
>
> See the "Paper Changes" update to all reviewers for more information, as well as discussion with other reviewers. Please let us know if we have addressed your concerns adequately.

---

> > ### Comment · Reviewer_RQ4u · 2024-12-02
> >
> > Thanks to the author for addressing my concerns. I am conflicted on how to update my assessment because while the answers to my questions are reasonable, the draft is missing the relevant performance numbers. However, in recognition of the novelty and value of this line of work, I will increase my score.

---

> ### Author Response · Authors · 2024-12-02
> **Replying**
>
> Hi, thank you for your reconsideration.
>
> For performance evaluation, we currently have tables 2 and 3 in Appendix D.1, as well as the entirety of Section 4 in the main paper. We added the new Appendix D.2 and Figure 10, which studies co-tenancy scenarios in which NDIF is servicing multiple NNsight requests. Which performance numbers is the draft still missing?
>
> We would be happy to add additional metrics for the camera-ready version.

---

### Public Comment · ~Neel_Nanda1 · 2024-11-15
**Paper Thoughts**

Chiming in as an uninvolved mechanistic interpretability researcher, I think that NDIF & NNsight have been highly valuable contributions to the interpretability community. I think this work is being somewhat underrated by the reviewers, and is highly worthy of acceptance to ICLR.

I consider NDIF the most important contribution. As clearly documented in the paper, there is a lack of work in academia on interpreting frontier language models - this makes sense, since hosting and running these is expensive and technically involved, and rarely makes sense for any given lab to invest in. NDIF enables these costs to be paid centrally, and to enable the work of many researchers. Most interpretability work is impossible without access to the model weights, so it wouldn't make sense for NDIF to try to cover proprietary models. But with highly capable open LLMs like LLaMA 405B, NDIF is able to significantly enable interpretability research on frontier LLMs that would otherwise not be practical. In addition to the scientific importance, as frontier LLMs become more widely used in the world, understanding them better seems vital, and certain projects can only be done on extremely capable models.

I also consider NNsight to be a significant contribution. The needs of interpretability research are fairly different from most ML work, and in my opinion good tooling is crucial. I've seen many projects made or broken by whether they had the right tooling. Speaking as the creator of TransformerLens, a similar library, I think the compiler + context manager style of NNSight introduces some valuable abstractions and enables new workflows. Even if most operations could be done by a skilled operator via PyTorch hooks, being able to do things intuitively, efficiently, and safely adds a lot of value (e.g. PyTorch hooks persist unless explicitly removed, which has introduced a *lot* of bugs into my code... A context manager elegantly solves this). Empirically, I've heard many researchers speak highly of NNsight for enabling their work.

This isn't to say that this paper doesn't also make valuable research contributions itself, by in my opinion enabling and improving further research is more important than how novel or insightful a work feels. It's much more impactful to make 20 good papers 20% better, than to write one good paper yourself. To quote the meta-review for [einops (ICLR 2022 Oral)](https://openreview.net/forum?id=oapKSVM2bcj):
> I would not have met [einops] had the paper not been submitted to ICLR, and hence I am certain it should be accepted, so more can see that we care not just about mathiness, but actually enabling progress in our field.
> The job of a conference like ICLR is to expose researchers and practitioners in machine learning to ideas and techniques that may advance their research and practice.

Einops is, of course, of interest to a much broader community than just interpretability, but I think the broader point still applies.

(Disclosure: I was not at all involved in this paper, but I do know the authors and likely have some positive bias due to that. No one asked me to write this)

(Note also: I don't feel confident in the norms here, so feel free to ignore this comment if these kinds of thoughts are not welcome! I don't see other researchers doing this, but I figure that as ICLR has made a deliberate choice to allow public comments during the rebuttal process, they likely want this kind of public feedback from uninvolved researchers, so long as it's constructive)

---

### Author Response · Authors · 2024-11-19
**To All Reviewers**

## Paper Changes
Thank you all for your insightful feedback on the paper’s contribution and presentation. Based on your feedback, we have made the following improvements to the PDF, and we welcome further comments, questions or suggestions:

1. To clarify how multiple concurrent user experiments are processed on NDIF, we have updated both Figure 5 and Section 3.3: Safe co-tenancy to discuss with more detail the meaning of co-tenancy and how we validate the correctness of NDIF’s co-tenancy features. (Reviewer RQ4u)
2. We have updated the related work section to include a discussion of S-LoRA and dLoRA (Reviewer RzZK).
3. We have updated the title to emphasize the fact that our system is built for open-weight models (Reviewers RQ4u and RzZK)
4. We have added a new Figure 8 to Appendix B, as well as a new "NDIF Design", Appendix B.2, to clarify how the NDIF system is implemented (Reviewers RzZK, RQ4u, JASr)
5. We have added a new "Load-Testing NDIF for Concurrent Users" Appendix section D.2 to address questions about stress-testing the system in this scenario (RzZK, RQ4u)

We respond below to two questions brought up by multiple reviewers.

## Access to closed, proprietary models (Rq4u, RzZK)
Two reviewers have cited the fact that our system covers exclusively open-weight models as a weakness. A prerequisite for creating a transparent API for closed models is to develop the technology demonstrating its feasibility. NDIF is the first step in this journey: our current focus is on open-weight models because commercial vendors will want to see the technical feasibility before they would consider a commercial deployment. Lines 526-533 in the paper expand the discussion on this topic.

## Experiment on reliability and Robustness (RzZK, RQ4u):
To address the points about evaluating NDIF for robustness as users increase, we evaluated the performance and scalability of the system as more users submit experiments.

We also respond to each individual reviewer in separate responses.

---

### Author Response · Authors · 2024-11-26
**To All Reviewers**

We thank you again for your work.

To address the points about evaluating NDIF for robustness as users increase, we have added a new Section D.2 in the appendix, where we measure the speed and robustness of NDIF with respect to increases in concurrent user count. Please see the new figure 9.

We also ask any reviewers that have not responded to our rebuttal to let us know whether we have addressed your comments adequately.

---

### Meta-Review · Area_Chair_FXR8 · 2024-12-11

**Metareview:**

The paper implements a system that abstracts common operations during ML experimentation on deep learning models, such as inspection of and intervention upon activations, collecting gradients, and modifying parameters. The abstraction allows experimental code to become portable and executable over more environments, especially distributed environments where maintaining custom model code is time-intensive and brittle.

Reviewers strongly agreed that the system solves an important problem for the research community. Initial reviewer concerns about the empirical validation and proof-of-real-world applicability were at least partly addressed. Reviewers agreed that, while there could be more experimental validation, the problem being tackled is very important to all ML researchers, while the technical approach is highly interesting and novel.

**Additional Comments On Reviewer Discussion:**

Reviewers were concerned about scalability and real-world use cases, which the authors have addressed through a real-world deployment case study on a 405B parameter model, though reviewers expressed they would have liked to see more such examples. One reviewer had concerns about the technical novelty of the proposed system's DAG design, which was discussed thoroughly with the authors and resolved.

---

### Decision · Program_Chairs · 2025-01-22

Accept (Poster)